**Infrasound and seismoacoustic signatures of the 28 September 2018 Sulawesi super shear**
**earthquake**
Christoph Pilger[1], Peter Gaebler[1], Lars Ceranna[1], Alexis Le Pichon[2], Julien Vergoz[2], Anna Perttu[3],
Dorianne Tailpied[3], Benoit Taisne[3]
1 – BGR (Federal Institute for Geosciences and Natural Resources), Hannover, Germany
2 – CEA, DAM, DIF, 91297 Arpajon, France
3 – EOS / NTU (Earth Observatory of Singapore / Nanyang Technological University), Singapore
Corresponding author: Christoph Pilger; BGR, Hannover, Germany; christoph.pilger@bgr.de
**Abstract**
A magnitude 7.5 earthquake occurred on 28 September 2018 at 10:02:43 UTC near the city of Palu on
the Indonesian island of Sulawesi. It was a shallow, strike-slip earthquake with a rupture extending to
length of about 150 km and reaching the surface. Moreover, this earthquake was identified as one of
very few events having a super shear rupture speed.
Clear and long-lasting infrasound signatures related to this event were observed by four infrasound
arrays of the International Monitoring System of the Comprehensive Nuclear-Test-Ban Treaty
Organization as well as by one national infrasound station in Singapore. Although these infrasound
stations SING (Singapore), I39PW (Palau), I07AU (Australia), I40PG (Papua New Guinea) and I30JP
(Japan) are located in large distances between 1800 km and 4500 km from the earthquake's epicentral
region, the observed infrasound signals associated to this event were intense, including both seismic
and acoustic arrivals.
A detailed study of the event-related infrasound observations and the potential infrasound generation
mechanisms is presented covering range-dependent infrasound transmission loss and propagation
modeling, characterization of the atmospheric background conditions as well as identification of the
regions of seismoacoustic activity by applying a back projection method from the infrasound receivers
to potential source regions. This back projection of infrasonic arrivals allows to estimate that the main
infrasound source region for the Sulawesi earthquake is related to the extended rupture zone and the
nearby topography. This estimation and the comparison to other super shear as well as large regional
earthquakes identifies no clear connection between the earthquake's super shear nature and the
strong infrasound emission.

**Keywords**
Infrasound; seismoacoustics;  propagation modeling; Sulawesi; super shear earthquake;

## 1. Introduction

Indonesia is located in a region with a very high rate of natural seismicity above a complex setting of plate tectonics. Subduction zones of convergent plate boundaries in this region define the largest faults of the Earth's crust, and the region of highest and most intense earthquake activity. In fact, some of the strongest and most destructive earthquakes recorded during the last decades have occurred in Indonesia, like the 2004 moment magnitude (Mw) 9.3 Sumatra-Andaman earthquake and various other events with Mw larger than 8 (*Pailoplee, 2017*). These strong offshore events can often generate large and devastating tsunamis. Additional crustal scale faults are also located on the Indonesian island of Sulawesi, including the Palu-Koro fault transecting the Northern part of the island (*Katili, 1978*). The frequent seismic activity associated to this fault was quantified using the United States Geological Survey (USGS) nearby seismicity data link (*USGS, 2018*), resulting in at least 60 earthquakes larger than magnitude 5 within the last 20 years and four events larger than magnitude 6 previous to the event discussed in this study.

The 28 September 2018 Sulawesi earthquake occurred at 10:02:43 UTC near the Indonesian city of Palu on the island of Sulawesi. It was estimated as a Mw 7.5 strike slip earthquake (*USGS, 2018*) along the Palu-Koru fault with a hypocenter location of 0.256°S and 119.846°E and a depth of about 20 km. Modeling indicates that the majority of the slip occurred shallow on the fault (above 10 km) with an offset of up to 7 m horizontal slip and a dip slip of up to only 2 m (*Socquet et al., 2019*). The rupture zone of the event extended north-to-south over roughly 150 km, along the fault and through the city of Palu, with a high rupture velocity of 4.1 km/s in average. This indicates it to be a so called super shear event having rupture velocities higher than the corresponding shear velocities (see *Bao et al., 2019; Socquet et al., 2019*). The phenomenon is comparable to the acoustic sonic boom, an effect where the source travels faster than its emitted waves. Analogous to acoustics the super shear rupture generates a shear wave mach cone, which may cause enhanced ground motion and result in increased damage potential (*Bernard and Baumont, 2005; Doan and Gary, 2009*). The Sulawesi earthquake resulted not only in intense ground shaking up to "considerable damages" of Modified Mercalli Intensity IX, but also in liquefaction, landslides, and local tsunamis within Palu bay (see *Heidarzadeh et al., 2019; Omira et al., 2019; Jamelot et al., 2019*). A large number of precursory earthquakes as well as aftershocks occurred surrounding this event.

The intense ground shaking of both the epicentral region and the topography nearby the Sulawesi earthquake resulted in strong and clearly observed infrasound signatures, which are the focus of this study. Infrasound, which is the sub-audible part of acoustic waves below 20 Hz, is generated by a large number of natural and anthropogenic sources (e.g. see *Le Pichon et al., 2010, 2019*) and can propagate over distances of thousands of kilometers with little attenuation to be recorded at highly sensitive infrasound arrays. Many sources of either explosive or eruptive characteristic, or those coming along with large mass movements can generate infrasound (e.g. *Gibbons et al., 2015a; Pilger et al., 2018*), including earthquakes.

Reports on infrasound from earthquakes in the USA (*Mutschlecner and Whitaker, 2005*) as well as in Peru, China and Chile (*Le Pichon et al., 2002, 2003, 2006*) indicate that the epicentral ground movement generates infrasonic pressure waves. Further studies on the Mw 9.3 Sumatra-Andaman earthquake (*Le Pichon et al., 2005*), the Mw 9.0 Tohoku earthquake (*Walker et al., 2013*) and on Italian earthquakes (*Marchetti et al., 2016; Shani-Kadmiel et al., 2017; Hernandez et al., 2018)* also highlight infrasound generated from tsunami waves hitting the coastline and from secondary phenomena like remote ground motion of mountain chains or extended basin areas. This secondary infrasound by remote ground motion is often called seismoacoustic waves, since the seismic waves (longitudinal, shear or surface) generated by an earthquake propagate to distant terrain features where the wave

energy is partly converted to atmospheric acoustic waves in the infrasound frequency range (e.g., see
*Arrowsmith et al., 2010; Hedlin et al., 2012*).
Although there are many studies about infrasound generated by earthquakes, only a small number of
earthquakes with a super shear rupture speed have been identified within the last 20 years (e.g.
Izmit/Turkey in 1999, see *Bouchon et al., 2000*; Kunlunshan/Tibet in 2001, see *Bouchon and Vallee,*
*2003;* Denali/Alaska in 2002, see *Dunham and Archuleta, 2004;* Quinghai/China in 2010, see *Wang and*
*Mori, 2012*; Craig/Alaska in 2013, see *Yue et al., 2013*), and only one publication known to the authors
identifies and investigates infrasound observations related to a super shear earthquake, namely the
Denali 2002 earthquake (*Olson et al, 2003*). Therefore, a main objective of this paper is to investigate
the potential of a connection between super shear earthquakes and infrasound recordings of large
amplitude.
This paper is structured as follows: Section 2 describes the data and methods applied within this study;
section 3 highlights the observations of epicentral infrasound and seismoacoustic signatures at remote
infrasound arrays; section 4 describes the modeling of infrasound transmission loss as well as
propagation and compares it to the observations; section 5 provides a back projection approach to
identify the acoustic source regions of the observed signals and discusses the event in comparison with
similar earthquakes.

**2. Data and Methods**

This study mainly considers data recorded at infrasound arrays of the International Monitoring System
(IMS, e.g. described in *Le Pichon et al., 2010, 2019*) established under the Comprehensive Nuclear-
Test-Ban Treaty (CTBT). The earthquake epicenter, as well as the nearest infrasound stations in
distances between 1800 km and 4500 km around the event, are shown in figure 1.

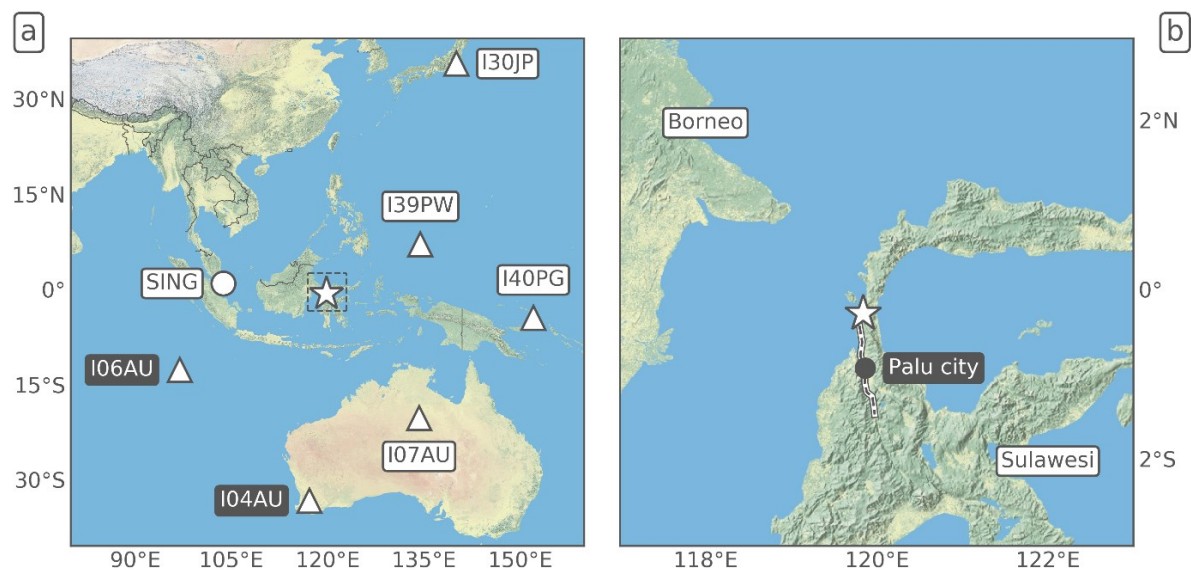

*Fig 1: a) Map of the Sulawesi earthquake epicenter (star) and the locations of the nearest surrounding*
*infrasound stations (the circle corresponds to a single-sensor station, the triangles to multi-sensor IMS*
*arrays; white-labeled stations registered the event, black-labeled ones did not). b) Zoom of the*
*epicentral source region showing in larger detail the rupture zone passing through the city of Palu.*

The two IMS infrasound stations closest to the earthquake epicenter clearly registered the event
(I39PW in Palau and I07AU in Northern Australia, see section 3). Two further IMS stations at larger
distances found clear indications of signals related to the earthquake (I40PG in Papua New Guinea and
I30JP in Japan, also see section 3). However, two other Australian stations (I04AU and I06AU) as well
as all of the more distant IMS infrasound arrays recorded no signals related to the earthquake source.
Additional data from a single infrasound sensor in Singapore (SING) was investigated and also showed
signatures related to the earthquake (see figure S1 of the supplement). However, due to a lack of array
calculations and directional information by only a single sensor, no further studies are applied for this
data.
The PMCC method (Progressive Multi-Channel Correlation, see *Cansi, 1995)* used in this study is
available from the DTK-GPMCC application in the NDC-In-A-Box package. The main objective of the
NDC-In-A-Box project is to offer to all National Data Centers (NDCs) of CTBT member states the
capability to process and analyse seismic, infrasound and hydroacoustic data, and so to become active
contributors to the verification regime of the CTBT. Technically, it consists of a number of automatic
and interactive software tools which are packaged in a Virtual Machine distributed by the CTBT
International Data Center (IDC). Among this set of software, DTK-GPMCC is the interactive array
processing tool, it allows to configure and run the PMCC detector from waveforms of any technology,
and to display and analyze the results. PMCC is applied to the raw differential pressure recordings at
each of the IMS infrasound arrays' microbarometers to derive advanced data parameters like back-
azimuth, apparent velocity and frequency content of coherent signals thereby associated to different
events (see figure 2). Back-azimuth reflects the horizontal direction of signal origin, while apparent
velocity indicates the arrival inclination, where higher values correspond to propagation from higher
altitude ducts. Signals are identified as pixel information in distinct time steps and frequency bands
and are clustered to signal families related to the same event. 1/3 octave band configurations with an
inverse frequency distributed window length are implemented between 0.01 and 4.4 Hz (*Garces,*
*2013*). Signals can be associated to a certain source by e.g. applying cross bearing techniques on the
back-azimuth directions of two or more arrays (*Matoza et al., 2017*). The seismic or acoustic origin as
well as the propagation path of signals, e.g. ducting via stratosphere or thermosphere (*Drob et al.,*
*2003*), can be inferred from the apparent velocity and frequency content of the recordings.
In order to further investigate and understand the infrasound detection pattern in the region following
the Sulawesi earthquake, various simulations were performed to compute acoustic transmission loss
and to simulate infrasound propagation between the source and the stations. Infrasound transmission
loss at surface level (see figure 3) was calculated using a frequency-dependent, semi-empirical
modeling technique coupled with realistic atmospheric specifications along the infrasound
propagation path (*Le Pichon et al., 2012; Tailpied et al., 2017*) in order to draw a range- and frequency-
dependent attenuation map estimating the acoustic pressure loss between source and receivers in
decibel (dB). The transmission loss of the signal at each station is associated to a confidence index that
integrates uncertainties from the propagation modeling and the atmospheric specifications.
Infrasound propagation (see figure 4) was modeled using a two-dimensional Parabolic Equation
method (NCPA PAPE, see *Waxler et al., 2017*) to quantify and visualize the ducting as well as amplitude
decrease between source and receivers.
In both the semi-empirical and the parabolic equation-based transmission loss estimates, data from
the European Centre for Medium-range Weather Forecast (ECMWF) meteorological model are used
to derive the effective sound speed as the most important background parameter for infrasound
propagation. Indeed, this parameter, defined as adiabatic sound speed modified by horizontal winds
in the propagation direction of the modeled sound, is used to provide the atmospheric background
conditions along the propagation path between the source and the stations (*Wilson, 2003*). Ducting
along tropospheric, stratospheric or thermospheric waveguides (*Drob et al., 2003*) can be estimated
in the same manner as the total amplitude loss from geometric spreading as well as atmospheric
attenuation (*Sutherland and Bass, 2004*). ECMWF values are used from 0 to 60 km altitude and merged
with temperature and wind climatologies above (MSISE00 and HWM07, see *Picone et al., 2002, Drob*
*et al., 2008*) to provide seamless effective sound speed profiles from 0 to 140 km altitude.
Back projection of the coherent earthquake-related signals observed at infrasound arrays to their
source region is performed within this study using a seismoacoustic method similar to that of *Marchetti*
*et al. (2016)* or *Shani-Kadmiel et al. (2017)*, which is also part of the built-in capabilities of PMCC (see
figure 5). Assumed is a conversion of the initial seismic wave with crustal propagation velocities of e.g.
4 km/s to acoustic waves with an average speed of e.g. 0.3 km/s at certain terrain features, like steep
or flat topography as e.g. mountain chains, islands, cliffs or extended plains. This method identifies the
seismoacoustic conversion areas and thus infrasonic source regions for the signals observed, taking
into account for each PMCC pixel the arrival time and back-azimuth direction relative to a point source
in space and time, here the Mw 7.5 earthquake epicenter. The cumulative sum and frequency of
occurrence of the backprojected origin locations therefore allows to identify infrasonic source regions,
either of epicentral or seismoacoustic origin.

## 3. Observations

The 28 September 2018 Sulawesi earthquake was identified in the recordings of four IMS infrasound
arrays: I39PW, I07AU, I40PG and I30JP. Four to six hours of differential pressure recordings from these
stations following the earthquake origin time (10:02:43 UTC) are analyzed using the PMCC method
described in section 2. Signal parameters related to the earthquake are extracted from the PMCC
results in terms of arrival time and duration as well as direction of origin (back-azimuth) and apparent
signal velocity.
These observation parameters for the four IMS infrasound arrays and for the earthquake-related signal
also identified in SING station data are summarized in table 1. Furthermore, source-to-station
distances as well as expected back-azimuth directions and arrival times using a celerity (epicentral
distance divided by the traveltime) of 300 m/s are presented for comparison. A graphical
representation of the waveform beams (bandpass-filtered between 0.6 and 4 Hz, except for I30JP,
where it is 0.02 and 0.1 Hz) and the main PMCC findings for the four IMS stations is provided in figure
2, highlighting epicentral infrasound arrivals and their acoustic characteristics in the observations but
also seismoacoustic and seismic signatures related to the event.

*Table 1: Findings from the observations of five infrasound stations and from theoretical distance-*
*azimuth calculations to the Sulawesi epicenter. Main signal groups are labeled with "IS" (infrasound)*
*and "SA" (seismoacoustic).*

| Station | SING | I39PW | I07AU | I40PG | I30JP |
|---|---|---|---|---|---|
| Distance to epicenter (km) | 1788 | 1845 | 2689 | 3604 | 4474 |
| Expected back-azimuth (°) | 94 | 243 | 322 | 276 | 213 |
| Expected 300 m/s arrival time (UTC) | 11:42 | 11:45 | 12:32 | 13:23 | 14:11 |
| Observed arrival time (UTC) | IS) 11:50 | IS) 11:36 SA) 12:34 | IS) 12:08 SA) 11:22 | IS) 13:05 SA) 12:37 | IS) 14:30 |
| Observed signal duration (min) | IS) 10 | IS) 25 SA) 7 | IS) 44 SA) 16 | IS) 24 SA) 8 | IS) 33 |
| Observed mean celerity (m/s) | IS) 267 | IS) 290 SA) 200 | IS) 304 SA) 514 | IS) 309 SA) 380 | IS) 263 |
| Observed mean back-azimuth (°) | - (no array) | IS) 251 SA) 257 | IS) 319 SA) 321 | IS) 275 SA) 276 | IS) 209 |
| Observed mean apparent velocity (m/s) | - (no array) | IS) 383 SA) 359 | IS) 356 SA) 371 | IS) 351 SA) 360 | IS) 436 |


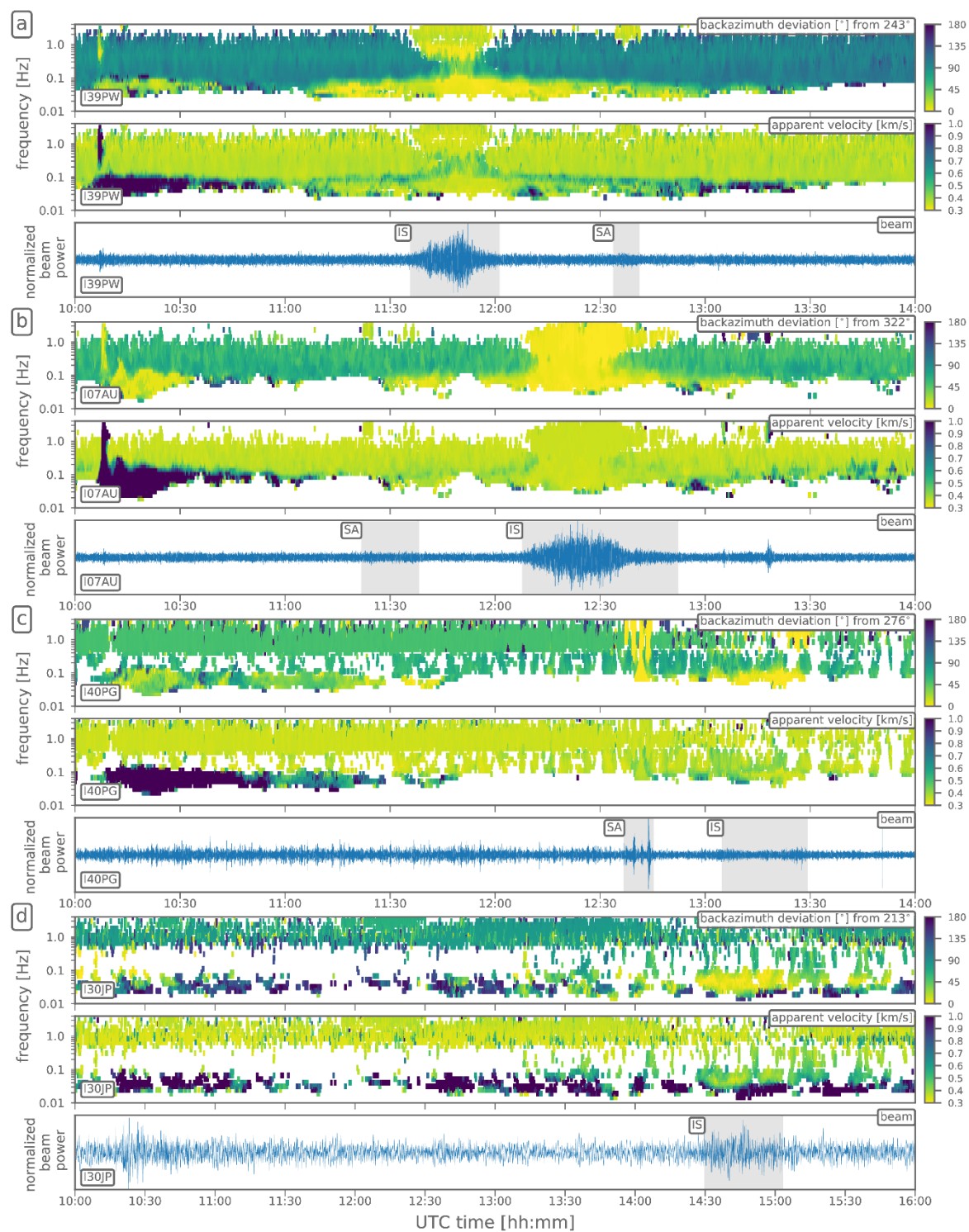


*Fig 2: Waveform beams and PMCC-derived results for the four infrasound arrays I39PW (a), I07AU (b),*
*I40PG (c) and I30JP (d; stations are ordered by epicentral distance, three frames per station, station*
*labels in the lower left corners). Shown in the corresponding stations' top frame are the observed back-*
*azimuth deviations from the direction to the earthquake epicenter (see labels in the upper right*
*corners), in the middle frame the observed apparent velocities, and in the bottom frame the waveform*
*beams. The whole 360° back-azimuth observations are converted to the given deviation plotting of ±*
*180°. Apparent velocities are saturated above 1 km/s. Beams are bandpass-filtered between 0.6 - 4 Hz*
*and four hours of data are shown with the exception of I30JP where the beam is bandpass filtered*
*between 0.02 - 0.1 Hz and six hours of data are shown. Main infrasound (IS) and seismoacoustic (SA)*
*signal groups (see table 1) are highlighted in grey.*

209

Initial seismic waves with high-frequency components (0.3-3 Hz) are found in I39PW and I07AU data arriving four to six minutes after the origin time, indicating apparent P-wave velocities of 4-10 km/s, lasting about two minutes. These are followed by low-frequency (0.05-0.5 Hz), quasi-continuous seismic waves observed in I39PW, I07AU, I40PG and possibly I30JP, likely related to seismic shear and surface waves, having velocities of 1-3 km/s. Aftershock activity as well as seismic signals from other regional earthquakes are also present in figure 2 for the hours after the main earthquake; aftershocks include 12 events of magnitude 5 or greater, and 40 events of magnitude 4 or greater within six hours following the event (*USGS, 2018*). Values for the arrival of seismic waves are not integrated in table 1, since the local microbarometer output generated from ground-shaking of the sensors are not the focus of this study. Nevertheless, the infrasound sensors do work fairly well as seismic arrays for this event (e.g. see *Gibbons et al., 2015b*) and the earthquake related seismic arrivals can clearly be identified in figure 2 having back-azimuths towards the epicenter and apparent velocities exceeding 1 km/s (drawn with dark blue colors in the middle frame plot of each station indicating seismic and not acoustic signal speeds).

Epicentral infrasound is clearly observed and produces the main signal with the largest waveform amplitudes in I39PW and I07AU (beams are plotted in figure 2 in the bottom frame plots of the respective stations, signals are highlighted by grey rectangles and "IS" labels). The analysis shows a broadband-frequency content (0.05 to 4.4 Hz) and long signal durations of 25 and 44 minutes (derived from the width of the high-frequency part signals originating from epicentral directions in the PMCC analyses). These signals are emphasized in figure 2, since the back-azimuth calculations as well as the array beams are focused towards the earthquake epicenter (yellow colors in the azimuth frame of each station indicating low to zero back-azimuth deviations from this direction). The low deviations from the theoretical back-azimuth directions (3° and 8°, see table 1 for the corresponding values) confirm the signals to be associated to either the epicenter, the rupture process at the surface or the ground shaking of topographic features on the island of Sulawesi. Crosswinds, as shown in figure S2 of the supplement, lead to certain back-azimuth deviations. An azimuthal sweep is observed in the I07AU data from south to north (directions of 316° to 323°), consistent with the north-to-south rupture along 150 km. Deviations from the expected backazimuth direction are largest in I39PW data (about ±10°). The other stations only show weak or no such variations. See figure S3 of the supplement for a detailed representation of these findings using absolute backazimuth values.

For the more distant stations I40PG and I30JP, the epicentral infrasound is consistent with the theoretical back-azimuths (1° and 4° deviation), but mostly allocated with frequencies below 0.1 Hz, indicating larger absorption of the high-frequencies along the long-distance propagation (see section 4 for the corresponding propagation modeling). The high-frequency pulses in the I40PG recordings around 12:40 UTC are associated to a seismoacoustic signal, which is discussed in the end of this section.

In general, the observed back-azimuths fit very well to the theoretical ones calculated for the epicenter for all four stations, allowing the application of a cumulative back projection method to locate the source regions of the observed infrasonic signals in section 5. The epicentral signals' mean apparent velocities are all in the acoustic range valid for stratospheric propagation (350 to 380 m/s, see table 1), with the exception of I30JP having higher mean apparent velocities of 436 m/s. This together with low celerity values of 263 m/s and appearance of only low-frequency signals at this station strongly indicates thermospheric propagation for I30JP instead of stratospheric. Thermospheric arrivals are expected to also be present in the other stations' observations apart from the dominant stratospheric ones; their later arrival time and lack of high-frequency content correspond to the long-lasting signal

families following the main signal peak for many minutes in the low frequencies. These signal families
can be observed together with low-frequency seismic wave activity and low frequency acoustic
components from the stratospheric ducting in frequency bands around 0.1 Hz. They are discernible
only to a certain degree by the apparent velocities and arrival times, being the slowest and latest
arrivals from the epicenter. The celerities observed at I39PW, I07AU and I40PG as well as the observed
arrival times and signal durations well correspond to the expected arrival times calculated using a 300
m/s celerity of average stratospheric propagation, quite close to the actually observed values at I39PW,
I07AU and I40PG (see table 1). The expected arrival times for these stations are clearly within the main
signals' observed time window and are only 2 to 6 minutes shifted from the respective mid-point of
the observed arrivals' time window (arrival time plus half of the signal duration).
Microbaroms, which are infrasonic signals from interacting ocean surface waves (*Donn and Naini,*
*1973; Ardhuin and Herbers, 2013*) are also present in the recordings of I39PW and I07AU around 0.2
Hz and dominant before and after the earthquake signals, as well as surf or potentially anthropogenic
noise in I40PG and I30JP data around 1 Hz during the complete observation. These background (noise)
signals can clearly be separated by back-azimuths (greenish colors in the top frame plots) from the
epicentral signal.
Seismoacoustic signals are identified in I07AU, I39PW and I40PG data, coming from nearly epicentral
directions and having acoustic apparent velocities. They have high frequency content (above 1 Hz) and
celerities below 200 or above 380 m/s, thus excluding purely acoustic waves propagating from the
epicenter at the time of the rupture, also those traveling through thermosphere or troposphere. These
signals could be seismoacoustic arrivals related to the earthquake (their signal parameters are
provided in table 1 and highlighted in figure 2 with the label "SA"). A conversion of seismic to acoustic
waves at certain, distinct terrain features might be responsible for this kind of signals. Islands between
Java and East Timor (south of Sulawesi) could be the rough source region of the I07AU and I39PW
signals, while islands of North Maluku (east of Sulawesi) may be the source of the seismoacoustic
signals in I40PG. Further details on backprojecting and thus identifying acoustic source regions are
provided in section 5.  Nevertheless, from the given observations it is not possible to certainly confirm
these signal locations as seismoacoustic source regions. None of the signatures are observed at more
than one station and smaller groups of signals come from all regions around Sulawesi, also including
neighboring islands like Borneo. These signals are not necessarily associated to the earthquake, they
could also originate from other local infrasound or ambient noise sources and are just coincidental to
the earthquake in direction and timing. Alternatively, they could be due to uncertainties in the array
processing or back projection methods.

**4. Modeling Results**
Transmission loss calculations using firstly a semi-empirical method for a horizontal representation
(map view, figure 3) and secondly a parabolic-equation-based propagation model for a vertical
representation (cross section, figure 4) are performed in this section to confirm and interpret the
observed epicentral infrasound signatures as described above. The semi-empirical method is used to
estimate the frequency-dependent transmission loss of a signal reaching the different infrasound
stations, thereby characterizing its detectability. Propagation modeling is necessary to identify
observed and expected signal arrivals, and to associate them to the prevailing atmospheric conditions
between source and receivers and the corresponding ducting behavior.
The quantification of infrasonic transmission loss is shown in figure 3 using the semi-empirical method
(see *Tailpied et al., 2017*) as well as quantifying of the stratospheric wind field in terms of intensity and

directionality. Simulations are performed within an 80° x 80° area using 0.5° x 0.5° spatial resolution around the earthquake epicenter for source frequencies of 0.2 Hz and 3 Hz. Most of the acoustic energy is concentrated at the low frequency band of 0.2 Hz. This was calculated applying the "The Infrasonic Energy, Nth Octave" (INFERNO) algorithm (see Garces, 2013) to the station data. It calculates acoustic energy with frequency bands based on the ANSI and ISO standards for noise characterization for the acoustic range extended into the infrasound range, and it is based on fractional octave bands. An example is shown in figure S1 of the supplement. Within this band the transmission loss calculated is similar for the closer stations SING, I39PW, I07AU and I40PG (see figure 3a) and their values are between 66 dB and 70 dB with uncertainties of about 4 dB (see table 2). While values at these four stations indicate a northwest-to-southeast corridor of signal amplitudes in the same order of magnitude, the other stations in northeastern and southwestern directions have slightly higher transmission loss values between 73 dB and 79 dB (see table 2), indicating less favorable ducting conditions and detection probabilities at these stations.

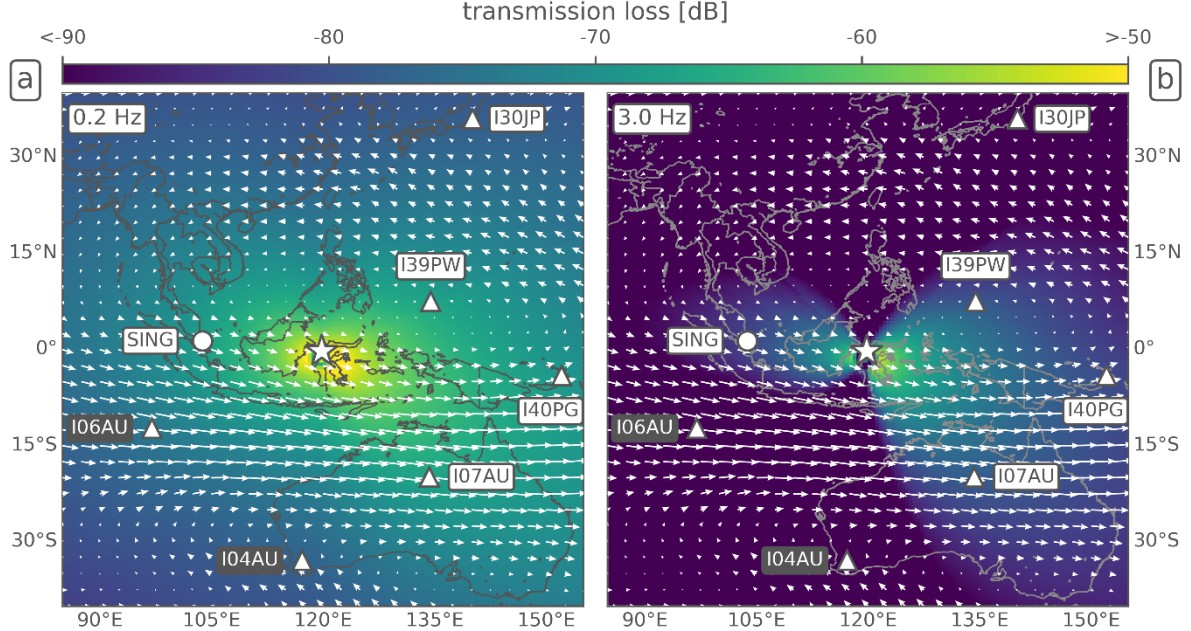

*Fig 3: Map quantifying the acoustic transmission loss in dB (color-coded), calculated for (a) 0.2 Hz and (b) 3 Hz source frequencies on a 0.5° x 0.5° grid. Arrows show direction and intensity of the stratospheric wind field averaged between 30 and 60 km altitude for the 28th of September 2018. The largest arrows represent a value of 25 m/s. For figure symbols and station labels see figure 1.*

*Table 2: Summary of transmission loss values and uncertainties (in dB), derived for all stations within this study from the frequency-dependent, semi-empirical method, as shown in figure 3.*

|  | SING | I39PW | I07AU | I40PG | I30JP | I06AU | I04AU |
|---|---|---|---|---|---|---|---|
| **0.2 Hz** | 69.3 ± 4.3 | 67.3 ± 4.4 | 66.8 ± 4.4 | 69.0 ± 4.3 | 78.2 ± 4.0 | 73.7 ± 4.2 | 77.3 ± 3.8 |
| **3.0 Hz** | 84.1±24.2 | 79.7±21.4 | 78.3±17.9 | 81.0±13.7 | 107.0±32.1 | 101.4±26.6 | 118.7±34.9 |

The similarity of the transmission loss values is consistent with the fact that low frequency signals are less affected by propagation effects along the path. Drawing the same picture with a source frequency of 3 Hz (figure 3b) indicates a different situation: station values for SING, I39PW, I07AU and I40PG now are between 78 dB and 85 dB with uncertainties of 13 to 25 dB (see table 2). These values are still quite similar to the ones estimated for 0.2 Hz, although the uncertainties for the calculation are

increased. The transmission loss calculated from the epicentral source into all directions to a stronger
degree visualizes for the high frequencies a focal effect in eastern and western directions with better
observation conditions, while having regions with increased transmission loss and thus more
unfavorable detection conditions in northern and southern directions. The stations' values in the
northern and southern directions are between 101 dB  and 119 dB with uncertainties of 26 to 35 dB,
indicating remarkably higher transmission loss for these three stations due to propagation effects and
atmospheric conditions and explaining, why no high-frequency signals (or signals at all) are observed
at the respective stations.
Stratospheric wind conditions affect the propagation especially for the higher frequencies and point
out the general possibility and effectiveness of a stratospheric duct. This is consistent with the fact that
high frequency signals are more sensitive to the atmospheric conditions along the propagation path,
also explaining the higher uncertainties in the calculation of these values. The stratospheric wind fields
shown in figure 3 support this sensitivity by estimating the direction of the dominant stratospheric
wind regime, which is eastward on the southern hemisphere's low latitudes, and the intensity of this
30 to 60 km average, which is up to values of 25 m/s. Strong tailwinds thus support the stratospheric
propagation to I07AU, while strong head- and crosswinds hamper it towards I04AU and I06AU. Winds
are weaker from the source towards the other stations, mostly due to the equatorial wind situation of
zonal stratospheric winds changing their direction here, rendering possible the simultaneous
propagation in western (SING), eastern (I39PW and I40PG) and to a certain degree probably even
northeastern directions (I30JP).
The given transmission loss modeling provides a map-based estimation at surface level where
stratospheric conditions are favorable or unfavorable for infrasound ducting. Complementary to this,
range-dependent propagation modeling is conducted between the epicenter and the four signal-
detecting IMS arrays to estimate the loss of signal amplitude due to atmospheric attenuation as well
as geometric spreading over the considerably large propagation distances of 1800 to 4500 km. This is
performed to estimate if stratospheric propagation is possible, even under weak ducting conditions or
conditions changing with distance.
The atmospheric ducting conditions and corresponding infrasound propagation for the four stations
are shown in figure 4. For I39PW, I07AU and I40PG, stratospheric ducting is modeled in good
agreement with the observed mean celerities of 290, 304 and 309 m/s (see table 1). Following *Negraru*
*et al. (2010),* celerities for stratospheric ducting are expected to be in the order of 280 m/s to 320 m/s.
Corresponding ray-tracing calculations (not shown here) estimate the celerities of those stratospheric
ducts to be  between 287 m/s and 293 m/s.
For I30JP, stratospheric ducting ceases along the 4500 km propagation path due to more unstable
ducting conditions and higher transmission loss (about 150 dB). This is also in good agreement with
the observations, since only a low-frequency signal is recorded at I30JP with a low celerity value of 263
m/s (ray-tracing suggesting 244 m/s), indicative not of a stratospheric but of a thermospheric arrival.

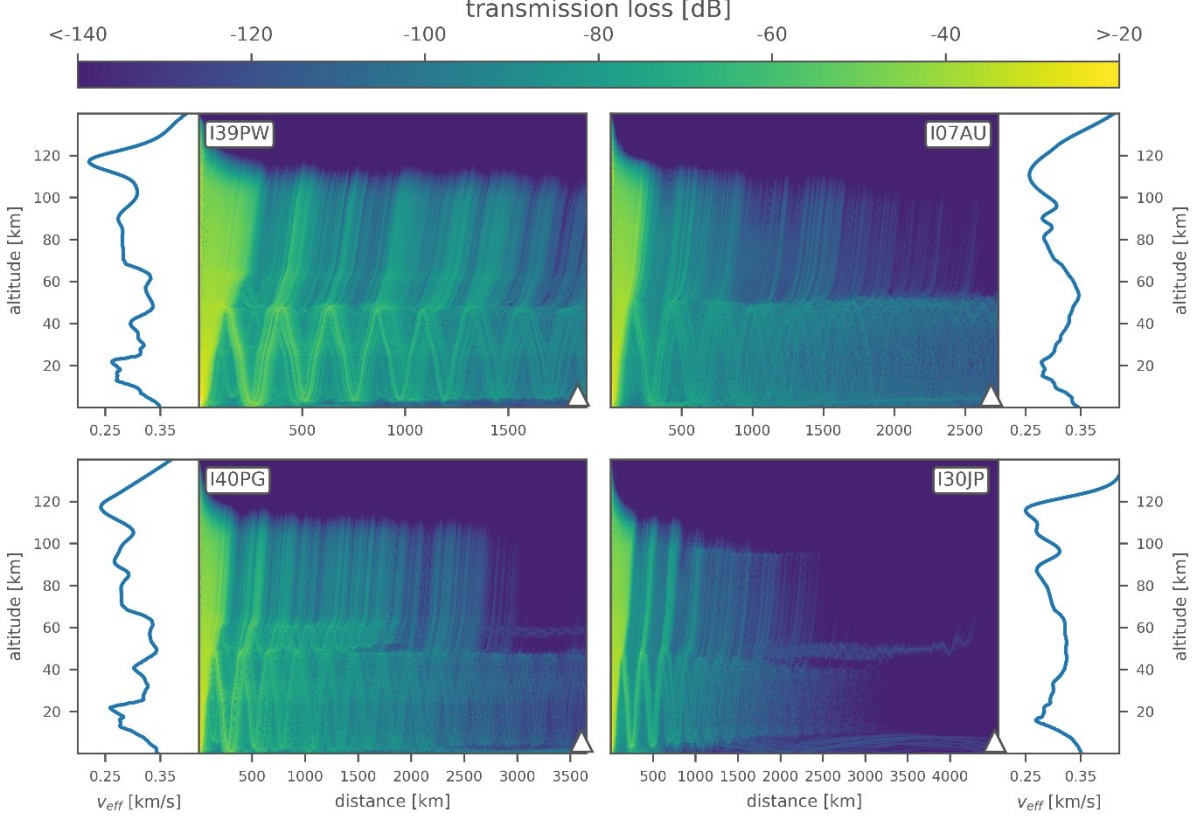


Fig 4: Propagation modeling between the Sulawesi earthquake epicenter (plot origins at 0 km distance) and the infrasound arrays I39PW, I07AU, I40PG and I30JP (respective triangles) using a range-dependent parabolic equation method, quantifying the transmission loss in dB relative to 1 km for a frequency of 1 Hz. An averaged effective sound speed profile ($v_{eff}$) is shown for each station.

Thermospheric ducts do not show up in figure 4, since this figure represents a 1 Hz modeling case highlighting the medium and high frequency stratospheric ducting and resulting in stronger absorption of thermospheric effects. For lower frequencies in the order of 0.01 Hz to 0.1 Hz, thermospheric attenuation is considerably small (*Sutherland and Bass, 2004)* and acoustic signal energy can propagate in the thermospheric duct over large distances with limited transmission loss.

The availability of atmospheric ducts can be quantified using the effective sound speed ($v_{eff}$) ratio between the stratospheric maximum (at 40-60 km) and the ground along the propagation path. This parameter indicates favorable ducting conditions, when being equal or larger than 1 and unfavorable conditions otherwise. Nevertheless, *Le Pichon et al., 2012* and *Landès et al, 2014* point out that also $v_{eff}$ ratios above 0.9 along the complete propagation path may lead to at least partially refracted energy in the stratosphere; whereas this ducting becomes highly likely for values above 0.95. While classical ray-trace modeling makes a strict separation between ratios larger or smaller than 1 (leading to existing or non-existing stratospheric ducts), the parabolic equation modeling used here also takes into account partial refractions of acoustic energy at effective sound speed ratios near but below 1. These partial refractions correspond to small-scale structures like atmospheric gravity waves, which vary the atmospheric temperature and winds and thus also influence infrasound propagation (*Kulichkov et al., 2010; Green et al., 2011*).

The $v_{eff}$ ratios of the average profiles depicted in figure 4 are 0.96 (I39PW), 1.00 (I07AU), 0.99 (I40PG)
and 0.93 (I30JP), fully supporting the reasoning above. Not shown in figure 4 are the propagation cases
to I06AU and I04AU, having no observations of the event and accordingly low $v_{eff}$ ratios of 0.92 and
0.93, while the propagation to the single element station SING is indicative of stratospheric ducting
with a higher $v_{eff}$ ratio of 0.98.

**5. Discussion and Conclusions**
The main focus of the discussion of observed and modeled signals from the 28 September 2018
Sulawesi earthquake is on the source regions and source mechanisms responsible for them. To support
this discussion, a back projection procedure (comparable to the one applied in *Shani-Kadmiel et al.,*
*2017* and in the supplement to *Gaebler et al., 2019*) is applied using the observed PMCC pixels and
backprojecting them using their temporal and directional information.
The back projection results towards the island of Sulawesi are presented in figure 5 in terms of an
event density map of the pixel-by-pixel information on their most likely origin locations. A total number
of about 107,000 pixels is used to derive the picture, combining the back projections of all four stations'
PMCC recordings towards the epicenter ±40° maximum deviation. Single station back projections can
be found in the supplementary figure S4. Seismic speeds of 4 km/s, resembling the primary
propagation of crustal seismic waves, are combined with 0.3 km/s acoustic celerities representing an
average value of the station observations. Uncertainties to the back-projected locations as seen by
extended contour regions in figure 5 are due to a number of potential influence factors. The choice of
a fixed seismic speed and fixed acoustic celerity for all pixels instead of individual values is supposed
to introduce location deviations. Measurement and analyses of back-azimuth directions may contain
uncertainties due to array configurations and due to crosswind influences on the infrasound
propagation. The method does not account for atmospheric variability (as does the forward
propagation approach of figure 4), introducing certain location biases. The velocity-averaged back
projection nevertheless sufficiently emphasizes the major source regions and infrasound generation
mechanisms.
A region to the south of the epicenter is highlighted (yellow colors representing the highest event
density), well corresponding with the earthquake rupture zone along the Palu-Koro fault line. Up to a
certain degree, this method also serves as a cross-bearing location procedure, although stations
contributing to it are not equally weighted but weighted by the number of pixels used from the
respective stations (in this picture, I07AU dominates the back projection, since it has the longest and
largest record of the event, also see figure S4); The location of the highest event density is at 119.6° E,
1.0° S, approximately 80 km south of the epicenter and thus half-way along the rupture.

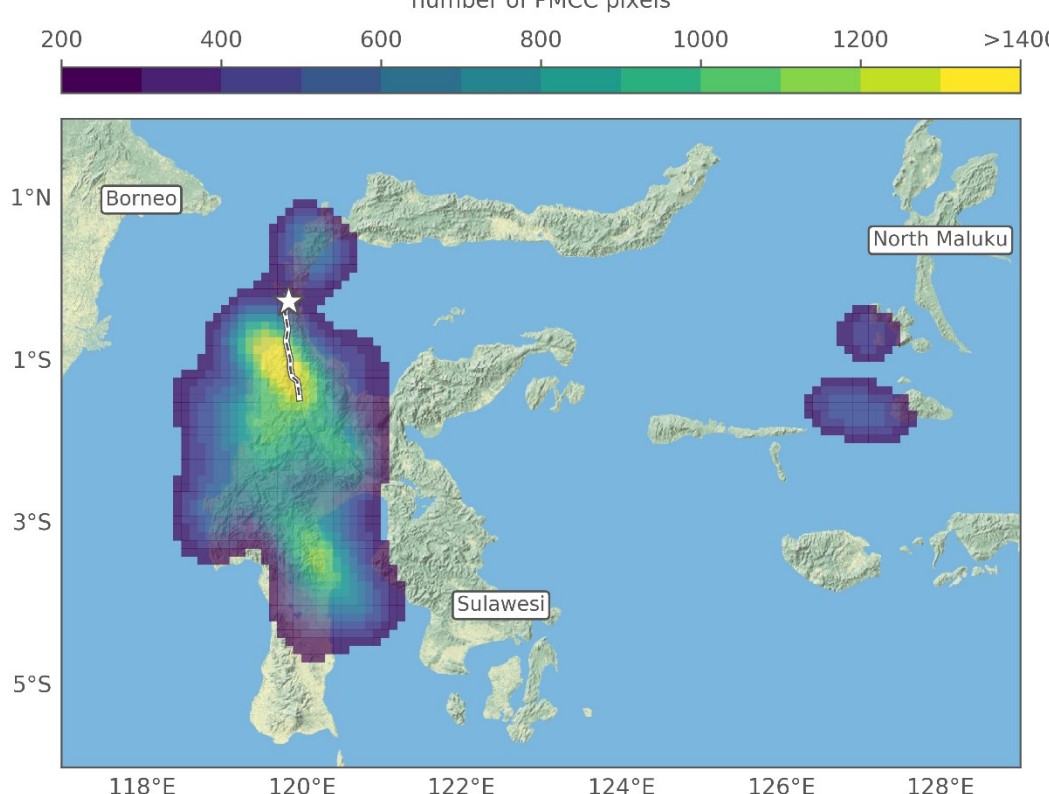


*Fig 5: Back projection of the combined PMCC detections from I39PW, I07AU, I40PG and I30JP.*
*Considered is each PMCC pixel's back-azimuth as well as a combination of 4 km/s seismic and 0.3 km/s*
*acoustic celerities, resulting in seismic-to-acoustic conversion locations. Color-coded event density for*
*these locations is shown on a 0.1° x 0.1° grid, highlighting regions with more than 200 backprojected*
*pixels per grid node. The epicenter is marked by an asterisk, the rupture zone traced by a dashed line.*

The figure highlights that infrasound is radiated not only from a distinct, epicentral point source alone, but from a region extended in north-south directions following the rupture (in fact the event density values at the epicenter itself are lower than those in the surrounding regions). Secondary peaks apart from the basin region around the rupture are identified north of the epicenter and in the southern part of Sulawesi island. The pixels of this southern secondary color peak are mostly related to the early parts of the main signal recorded at I07AU, while the central and northern color peaks in the figure are related to signals arriving some minutes later. This corresponds to the 316° to 323° sweep in I07AU data from south to north, as described in section 3. The two side-maxima separated from the main signal's colored region are related to the seismoacoustic signatures described in section 3. They are derived from a number of I40PG PMCC pixels and point to a region near the North Maluku islands east of Sulawesi (also see figure S4). Other side-maxima as e.g. the ones between Java and east Timor, also mentioned in section 3, are beyond the map borders and not shown here, but can be found in figure S4.

In general, the results observed and visualized by figure 5 point out that an enlarged region, closely following the rupture and thus also the topography along the fault, generates the acoustic signals recorded at the remote infrasound sensors. This includes the rupture region itself suffering most from the earthquake-related ground motion (offsets of up to 7 m horizontal and 2 m dip slip) as well as an extended basin area around the rupture, enclosed by mountain chains in mostly north-to-south directions. Mountainous areas are a well-known source of seismoacoustic signatures (e.g. *Arrowsmith*

*et al., 2010*), and correspond to the event density maxima in figure 5: the mountain chains west and
east of the Palu-Koro fault as well as the mountain area in the south of the island with the highest
mountains of the Sulawesi island (Mt. Rantemario and Mt. Rantekombola, both about 3500 m
elevation) generate large portions of the recorded signals. The less prominent but recognizable regions
north of the epicenter (Mt. Fuyul Sojol, 3000 m elevation) and on the Maluku islands (e.g. Mt. Buku
Sibela, 2000 m elevation) are also related to topographic peaks. The most likely source mechanism for
the generation of large parts of the seismoacoustic signals is therefore estimated to be the shaking of
elevated or exposed topography, stimulated by crustal seismic or surface waves reaching these areas
and turning them into motion.
To qualitatively assess if the super shear nature of the given earthquake or the regional prerequisites
(or both) are responsible for the intense and long-lasting infrasound signals observed, the 28
September 2018 Sulawesi earthquake is compared to three other super shear earthquakes as well as
three other normal shear earthquakes from the same region (Indonesia and Papua New Guinea).
Shallow events between 5 - 30 km depth were chosen with comparably strong magnitudes of Mw >6.5
so that infrasound generation and detection can be expected. Table 3 chronologically lists these six
events and provides an estimation of the emitted and observed infrasound for all of them.

*Table 3: List of events similar to the 28 September 2018 Sulawesi earthquake, either in their super shear*
*nature or in their regional origin. The separation between "Event detection" / "No Event Detection" is*
*an estimation following data analyses performed by authors of this study.*

| Event (with location, time, magnitude, depth) | Source type | Available IMS stations up to 5000 km distance | |
|---|---|---|---|
| | | **Event Detection** (with distance and PMCC-estimated signal duration) | **No Event Detection** (with distance) |
| **Denali,** Alaska/USA, 03.11.2002, Mw 7.9, depth 4.9 km | Super shear earthquake | I53US (156 km, 10 min) I10CA (3358 km, 30 min) | I59US (4919 km) |
| **Sumatra Andaman,** Indonesia, 26.12.2004, Mw 9.3, depth 30 km | Same region, normal shear earthquake | I52GB (2852 km, 30 min) | I07AU (4930 km) |
| **Quinghai,** China, 13.04.2010, Mw 6.9, depth 17 km | Super shear earthquake | I34MN (1810 km, 10 min) | I46RU (2480 km) I45RU (3273 km) I31KZ (3669 km) I30JP (3996 km) I39PW (4831 km) |
| **Craig,** Alaska/USA, 05.01.2013, Mw 7.5, depth 10 km | Super shear earthquake | I53US (1294 km, 5 min) I56US (1443 km, 10 min) | I10CA (2647 km) I57US (2795 km) I18DK (3509 km) I44RU (4236 km) I59US (4334 km) |

| **Porgera,** Papua New Guinea, 25.02.2018, Mw 7.5, depth 25.2 km | Same region, normal shear earthquake | I40PG (1044 km, 60 min) I39PW (1759 km, 45 min) I07AU (1784 km, 45 min) I60US (3835 km, 45 min) I04AU (4164 km, 15 min) | I22FR (3144 km) I05AU (4064 km) I30JP (4587 km) |
|---|---|---|---|
| **Kokopo,** Papua New Guinea, 14.05.2019, Mw 7.5, depth 10 km | Same region, normal shear earthquake | I40PG (72 km, 10 min) I39PW (2379 km, 30 min) I22FR (2527 km, 10 min) | I07AU (2649 km) I60US (3004 km) I05AU (4286 km) I30JP (4542 km) I58US (4803 km) |


The three super shear earthquakes named after the Denali fault, the Quinghai province and the city of Craig, occurring in 2002, 2010 and 2013, are the earthquakes most recent, most intense and most similar in their super shear characteristics to the 28 September 2018 Sulawesi earthquake, also having super shear rupture velocities of 4 to 6 km/s (see *Dunham and Archuleta, 2004; Wang and Mori, 2012; Yue et al., 2013*). Although the IMS infrasound network is not fully established yet (to the time of the Sulawesi earthquake, 80% of the stations were certified and operational, while it were only 8% to the time of the Denali earthquake and about 70% during the time of the other two earthquakes), at least one infrasound array was able to unambiguously detect and characterize each of the mentioned earthquakes.

The infrasound signals for Denali earthquake indicate a high signal-to-noise ratio at the nearby I53US station as well as a much weaker signal at I10CA in a much larger distance. This event was a good opportunity to track the infrasound back to its generation region in the Alaska Mountain Range along the Denali fault where the rupture occurred (observed in I53US data, *Olsen et al., 2003*) and to the Rocky Mountain Chain south-east of it (observed in I10CA data), where similar observations were made for the 1964 Great Alaskan earthquake (see *Young and Greene, 1982*). The strong movement of local and remote topography generated the infrasound in good agreement with the Sulawesi case. However, no indication is given that the super shear characteristics of the Denali earthquake specially favors the generation of infrasound. For the Quinghai and Craig earthquakes, also reported to be super shear, much weaker and shorter duration infrasound is observed at stations in distances of 1400 km (I53US to Craig) to 1800 km (I34MN to Quinghai), compared to Sulawesi where stronger and much longer infrasound signals were observed between 1800 km and 4500 km. Again, these do not indicate any connection between those previous super shear earthquakes and extraordinary infrasound generation.

The Sulawesi earthquake is also compared to three strong earthquakes within the same region, most prominently two nearby Papua New Guinea earthquakes (near the Porgera mine, 2018 and Kokopo city, 2019) of the same magnitude occurring half a year before and after the Sulawesi one, showing clearly observed infrasound signals with high signal-to-noise ratios at multiple IMS stations as well. These infrasound signals are observed up to similar distances as in the Sulawesi case and also provide long-duration, strong amplitude wave energy associated to infrasonic and seismoacoustic arrivals coming from the two earthquakes. Clear seismic signals are also present in the recordings (as in most cases described before, apart from Quinghai) and an association to topographic features as infrasound source regions is possible (the mountain chain in central Papua New Guinea for Porgera and the mountain areas in New Britain and New Ireland for Kokopo). For the Sumatra Andaman earthquake of 2004, strong infrasound with long signal durations was observed and could be backprojected to topographic features of islands and shorelines, especially where the follow-up tsunami reached the

shoreline of the Bay of Bengal (see *Le Pichon et al., 2005*). None of the presented earthquakes were
super shear earthquakes, but all of them, especially the two very similar Papua New Guinea
earthquakes generated strong infrasonic signals comparable to the signals of the Sulawesi event.
It can be concluded from comparison with other events above that strong infrasound generated by an
earthquake is not mainly or exclusively linked to the earthquake's super shear characteristic, but most
likely to the nearby existence of mountainous topography. This topography serves as a large-area
resonating membrane in terms of large masses brought into motion by a triggering earthquake. These
mass movements produce large amounts of acoustic energy, which can be recorded at nearby or
remote infrasound stations given conducive propagation conditions.
The given super shear event resembles one of only few large magnitude, shallow earthquakes
generating pronounced infrasound. It therefore provides a unique opportunity to study earthquake
generated infrasound in terms of the source mechanisms, signal characteristics, propagation
conditions and ducting behavior. It also supports the improved understanding of the process of
infrasound radiation by mountain shaking from large earthquakes and the conversion of seismic to
acoustic energy.
Measurement uncertainties within this study are due to the instrumentation and methods applied;
modeling uncertainties are due to assumptions applied within the models and due to multi-scale
atmospheric variations between source and receivers leading to uncertainties in the transmission loss
and propagation calculations. Taking into account these uncertainties allows to improve the methods
and models to cope with such issues in the future. It will help to gain novel and enhanced insights
about infrasound observations and modeling in general and earthquake generated infrasound in
particular. This will also help to optimize seismoacoustic observation networks in terms of better
understanding the instrumental needs and better evaluating the signatures observable by it. It will
finally support seismoacoustic studies of natural as well as anthropogenic infrasound sources in the
future and thereby support the infrasound monitoring for treaty verification purposes of the CTBT, as
did other CTBT-related studies about infrasound observation, propagation and signal characterization
(Assink et al., 2016; Bowman, 2019; Gaebler et al., 2019).

**Acknowledgements**
This work comprises Earth Observatory of Singapore contribution no. 249. This research is partly
supported by the National Research Foundation Singapore and the Singapore Ministry of Education
under the Research Centres of Excellence initiative.

**Data availability**
Information about earthquake magnitude, location and frequency of occurrence in the region of
interest is retrieved from the online-accessible archive of the USGS, see
https://earthquake.usgs.gov/earthquakes/ (last accessed 02.09.2019).
Atmospheric wind and temperature profiles are derived from operational high-resolution atmospheric
model analysis, defined by the Integrated Forecast System of the ECMWF, available at
https://www.ecmwf.int/ (last accessed 02.09.2019).
Waveform data for the infrasound arrays of the CTBTO IMS (https://www.ctbto.org/) used in this study
are available to the authors being members of National Data Centers for the CTBTO. Waveform data
for SING infrasound station are available to the authors being members of the Earth Observatory of
Singapore.

**Competing Interests**
none

**Author Contributions**
**CP** analyzed the waveform data, performed the propagation modeling, wrote the manuscript text and
coordinated the co-author contributions; **PG** compiled the data, generated the figures and helped with
finalizing the manuscript layout; **LC** provided first ideas and initiated the collaborative study; **ALP**
provided expertise in earthquake infrasound, comparison to other events and initiated the
collaborative study; **JV** analyzed the waveform data and performed propagation modeling; **AP**
analyzed the waveform data and provided manuscript text; **DT** performed the attenuation modeling
and provided manuscript text; **BT** provided first ideas and initiated the collaborative study; **all authors**
supported and improved the draft by proof-reading, commenting or correcting the manuscript.

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

**Supplementary Material**

**Figure S1**

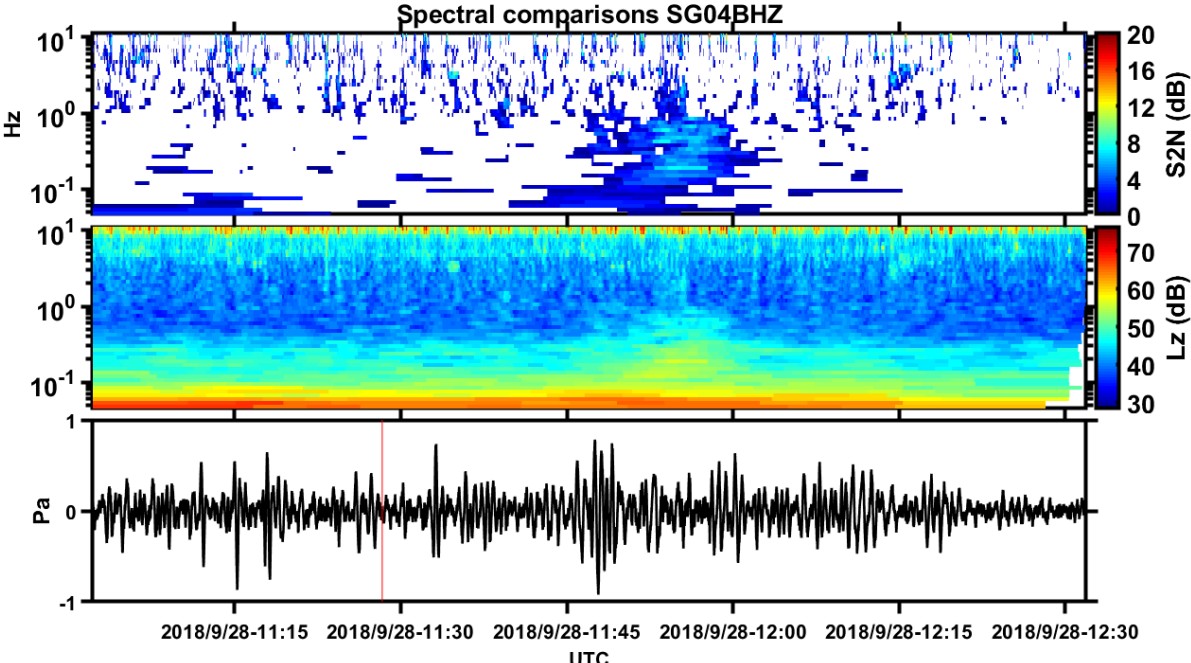


*Fig S1: Example of INFERNO analysis for the Singapore station (SING). The bottom panel is the*
*waveform recorded at sensor SG04, with the red vertical line representing an estimated arrival time*
*based on the location and time of the earthquake. The middle panel is an example of an INFERNO*
*spectrogram where energy is calculated in fractional octave bands. The top panel is a signal to noise*
*plot derived from the spectrogram. All the values for each frequency band are averaged and a 3dB*
*threshold is set. Note that while the signal from the event is not as obvious within the waveform and*
*spectrogram, the signal to noise plot clearly shows the signals arrival.*
**Figure S2**

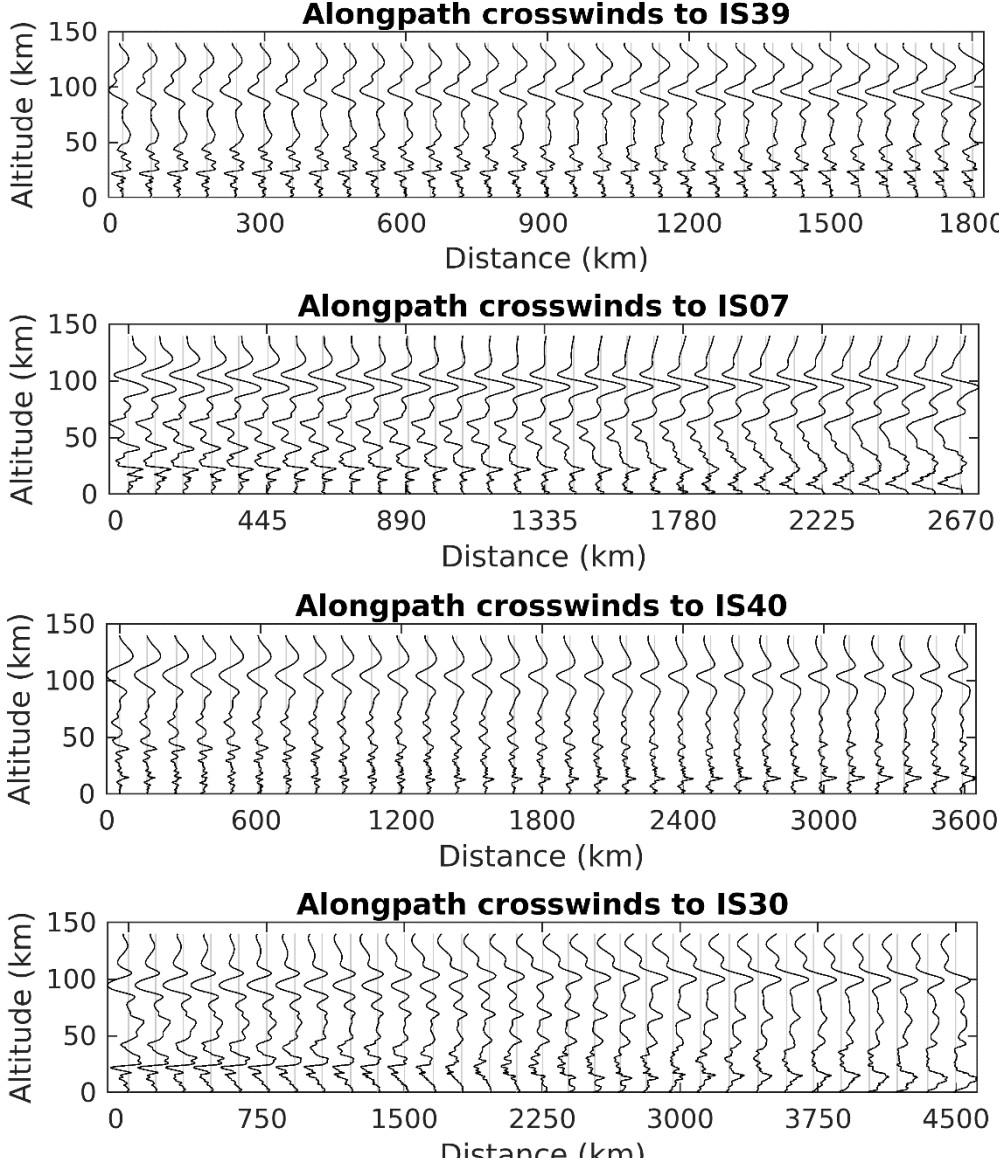

*Fig S2: Crosswind profiles along the propagation path from the epicenter to the four arrays. Positive*
*values correspond to winds in the 90° clockwise perpendicular direction, the distance between two*
*vertical lines corresponds to 50 m/s wind intensity. Range-dependent ECMWF profiles merged with*
*climatologies are used as described in the data section. Enhanced positive crosswinds potentially*
*responsible for positive back-azimuth deviations occur at I39PW around the stratospheric turning*
*altitude of 50 km and to some degree below that altitude. Strong negative crosswinds at 50 km altitude*
*and below occur at I07AU and might explain negative back-azimuth deviations for this station. Weak*
*total crosswinds at and below 50 km at station I40PG might explain marginal back-azimuth deviations*
*at this station. Strong thermospheric crosswinds around 100 km and below might explain back-azimuth*
*deviations at I30JP after thermospheric propagation.*
**Figure S3**

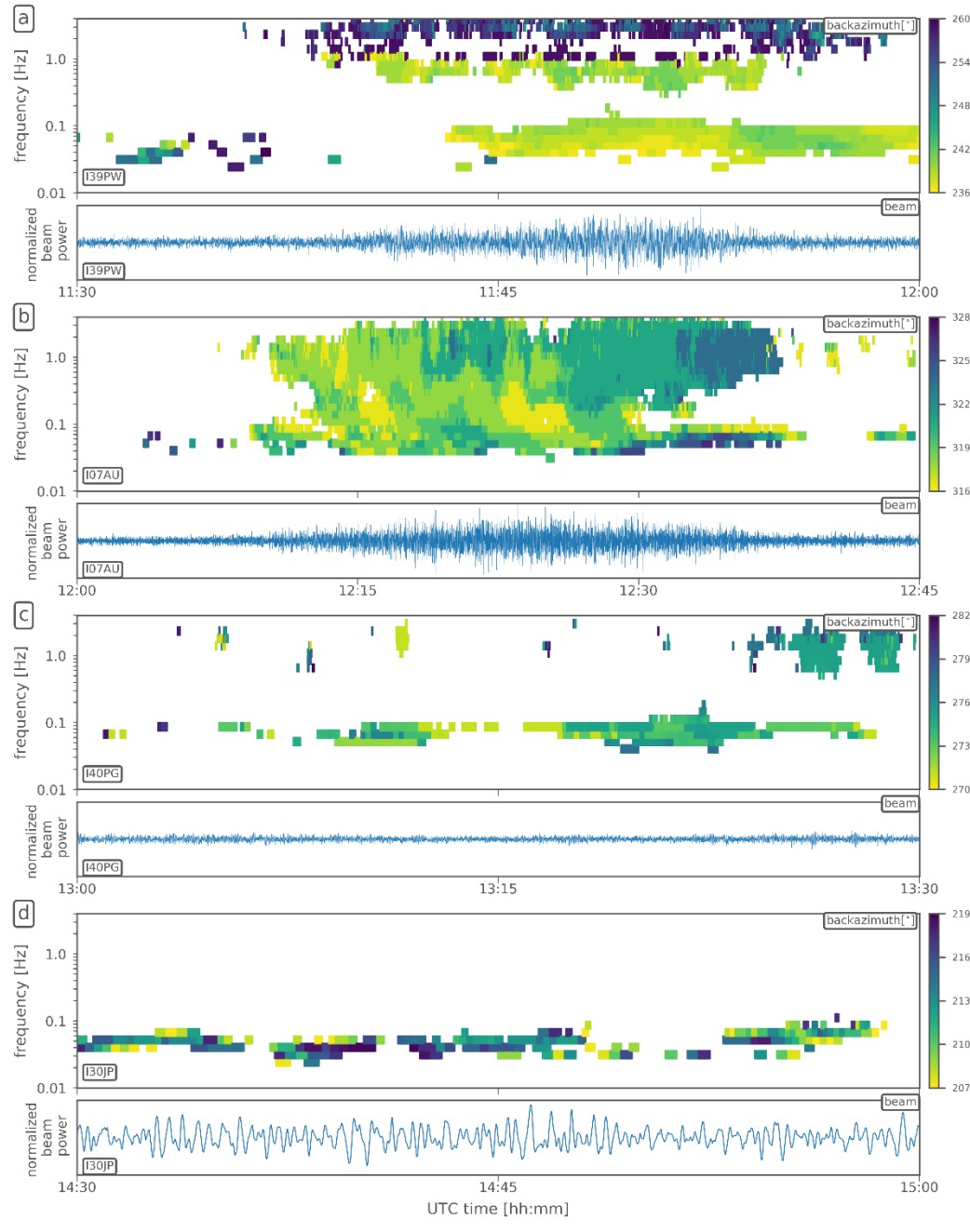


*Fig S3: Waveform beams and PMCC backazimuth information for the four infrasound arrays of figure*
*2. Absolute backazimuth directions are provided here instead of epicentral deviations, allowing to*
*quantify changes and differences in the direction of signal origin. A small azimuthal section (±12° from*
*the expected epicenter direction in subfigure a, ±6° in subfigure b, c and d) and short time window (30*
*min in subfigures a, c and d, 45 min in subfigure b) is chosen to highlight the epicentral infrasound's*
*origin direction and arrival time as specified in table 1. Differences in the direction of origin between*
*the high-frequency and the mid- to low-frequency parts of the epicentral infrasound are found at I39PW*
*(subfigure a) in the order of ±10°. An azimuthal sweep of about 7° is observed at I07AU (subfigure b).*
*Both phenomena indicate a spatially and temporally extended source. Only small and mostly arbitrary*
*backazimuth variation is present at stations I40PG and I30JP (subfigures c and d).*
**Figure S4**



*Fig S4: Single-station back projection maps for the four infrasound arrays. The colorbar of each figure*
*starts at a 4 time lower value than in the cumulative 4-station-figure 5. The main region near the*
*epicenter (marked by a star) and the rupture south of it are projected reasonably well for each station.*
*The directional deviation and spatial extension of the back-projected source regions per station*
*corresponds to backazimuth variations as e.g. the azimuthal sweep in figure S3. Additionally, regions*
*of potential seismoacoustic signal generation are identified around the island of Sulawesi, as described*
*in the manuscript text.*