# Peer review of "Infrasound and seismoacoustic signatures of the September 28th 2018 Sulawesi super shear earthquake 3 4 5 Christoph Pilger1, Peter Gaebler1, Lars Ceranna1, Alexis Le Pichon2, Julien Vergoz2, Anna Perttu3, 6 Dorianne Tailpied3"

_Natural Hazards and Earth System Sciences, 2019_

## Referee Comment (RC1) · Sven Peter Näsholm (Referee) · 11 Jul 2019

**Reviewer feedback**

| | |
|---|---|
| Journal: | Natural Hazards and Earth System Sciences Discussions (NHESSD) |
| Manuscript ID: | nhess-2019-179, submitted by the authors on 5 June 2019 |
| Manuscript title: | *Infrasound and seismoacoustic signatures of the September 28th 2018 Sulawesi super shear earthquake* |
| Authors: | Christoph Pilger, Peter Gaebler, Lars Ceranna, Alexis Le Pichon, Julien Vergoz, Anna Perttu, Dorianne Tailpied, and Benoit Taisne |
| Handling Editor: | Maria Ana Baptista |
| Invited Reviewer: | Sven Peter Näsholm, NORSAR, Norway |

**SUMMARY AND RECOMMENDATION**

Pilger et al., provide a data analysis on atmospheric infrasound related to the Sulawesi 28 September earthquake as recorded by a set of ground-based stations in Asia and Oceania. The processed infrasound waveform analysis is supported by transmission loss estimates based on parabolic equation-based propagation modelling, and on a semi-empirical formula developed in previous works. The key physical mechanism for generation of these infrasonic acoustical pulses is assumed to be ground-to-atmosphere coupling stimulated by the earthquake-related vibrations.

The authors also map the geographical regions where infrasound is generated using a back-projection approach which utilizes the wavefront backazimuth of arrivals at the stations under the assumption of 300 m/s acoustical wave celerity and 4 m/s solid earth wave speed. This leads to the conclusion that the earthquake main rupture zone and topography nearby are the key regions for infrasound generation.

Moreover, a comparison with other events suggests that the strong infrasound observed for the studied event is not related to the super-shear nature of the earthquake.

**Conclusion**

I think a manuscript revision will be necessary before this discussion paper can be accepted as an NHESS research article. I look forward to reading the updated manuscript and foresee it to be straightforward for the authors to assess the recommendations given and to implement the relevant edits.

**Recommendations on specific aspects**

1) Super-sherar related analyses:

   I recommend providing a more concise background on why it is interesting to understand if the observed strong infrasound waves are related to the super-shear nature of the earthquake. Then the discussion and comparison with other earthquakes can be linked to the added background information. For example, it could be explained more precisely what is a super-shear earthquake and it could be mentioned that these

can be associated with an effect analogous to acoustic sonic booms with the source traveling faster than the wave propagation speed.

Since the detectability is naturally influenced by the presence of infrasound arrays within the waveguides available from the source, the authors could consider complementing Table 2 with additional information on the infrasound array network available around each of the listed events.
What are the criteria used for the assessments *weak infrasound* and *strong infrasound* in Table 2? I would also recommend stating the distance between event and station, instead of no or vague information about distance (*nearby / remote*).

It would be informative with a concise explanation why the registrations (or non-registrations) of seismic arrivals on the infrasound stations are relevant to the assessment? (Otherwise, I would recommend removing this information from Table 2.)

2) Back-projection analysis:

I would recommend reinforcing the discussion on the back-projection analysis. Based on the current displays, I find it hard to analyze what are the key wavetrain segments on the different stations that contribute to the different highlighted geographical regions shown in Fig. 5.

Maybe it would be useful to provide an additional back-projection result map per station, either in the paper body or in an appendix? This might make it more straightforward to the reader to see, e.g., if the two blue colored source regions at around 127°E are related to signals on one or on multiple stations. This could also help providing insight in the way each of the stations constrains the source region estimates on Sulawesi. As a suggestion, the coloring of the added per-station maps might be a function of the reduced time for the corresponding infrasound detections? This could then allow for an interpretation of links between array parameter output as a function of time (and the associated waveforms) in the context of the different source regions and features.

A separate back-projection image per station would also allow the reader to better follow and assess the discussion in the last paragraph of Section 3.

3) Suggested references to add:

- Studies including back-projection of infrasound recordings: Assink et al. (2018); Walker et al. (2013).

- A study which shows observations of infrasonic waves generated by seismic surface waves along the Rocky Mountains: Young and Greene (1982).

4) Atmospheric model product:

From which ECMWF atmospheric model product are the wind and temperature fields extracted?

5) Transmission loss:

The transmission loss is estimated both using parabolic equation simulations in a lossy medium, and using a semi-empirical formula. However, I find a potential confusion in the mixed use of the concepts of attenuation and transmission loss in the text. Figure 3 and Figure 4 display the results of transmission loss estimates using the two techniques. If you agree on this viewpoint, I propose to streamline the language and

the discussion of the associated results to underline that both approaches are applied to calculate transmission loss.

For example, line 137 can be confusing. It could be modified from:
*In both attenuation and propagation modeling, data from the European . . .*
into:
*In both the semi-empirical and the parabolic equation-based transmission loss estimates, data from the European . . . .*

Moreover, it would be interesting to see a direct comparison between the two transmission loss estimates as calculated for the station sites.

6) Nomenclature:

- I suggest homogenizing the wording related to the concept of *back-projection*. The paper also uses *backtracking* and *back-tracking*, while some other works in addition consider a more general concept of *back-propagation*. If the mix is intentional, a more concise explanation is needed. Otherwise, I suggest using only one of the constructs in the text.

- The paper could be clearer and more consequent on the use of the different physical infrasound and seismic wave generation processes considered in observations, analyses, and figure labels. Currently, the constructs *epicentral infrasound; seismoacoustic signals; secondary infrasound; seismoacoustic precursors; seismoacoustic successors; secondary signals* are used, and I think the readers would appreciate a consolidated use and definition of these concepts.
I'm not sure whether the definition of *seismoacoustic waves* provided on Line 79 is appropriate:
*. . . also highlight infrasound generated from secondary phenomena like remote ground motion of mountain chains or extended basin areas, and from tsunami waves hitting the coastline. This secondary infrasound is often called seismoacoustic waves, . . .*
Is epicentral infrasound supposed to be included in your definition of a seismoacoustic waves?

**Specific text edit suggestions**

1) Line 107:
SING data is not shown, which makes it difficult to assess the summary given on these data signatures. I propose including a SING signal plot either in the paper body or in an appendix.

2) Line 116:
I would advise including a reference or explanation to describe what is an NDC and what is the NDC-in-a-box software.

3) Line 124:
It needs to be better clarified what is meant when claiming that *the propagation of signals* can be identified from the apparent velocity and frequency content.

I assume this would be related to identification of the atmospheric ducts penetrated by the acoustic waves (tropospheric / stratospheric / mesospheric / lower thermospheric?

4) Line 130:

I suggest including a concise statement of the parameters that go into the calculation, preferably complemented by the full equation for the semi-empirical transmission loss estimates. Is this equation 2 in Le Pichon et al. (2012)?

5) Line 165:

How is the signal duration estimated?

6) Line 168:

I'm not sure celerity is best described as *speed over ground*? Maybe you could use, e.g., *ratio of the range to the traveltime*, or *epicentral distance divided by the traveltime*?

7) Line 178 (Figure 2):

- Would a separation of this figure into four labeled subfigures (one per station) make the grouping of the subpanels more apparent? I think the readability of Figures 2–4 would also benefit from using a larger text label font (or larger figures).

- The middle panel labels for each station say *signal speed*, while the paper text otherwise uses *apparent velocity*. I think this should be consolidated.

- Suggested clarification: *are sorted by distance from above* $\longrightarrow$ *are ordered by epicentral distance.*

- The caption should explain what the labels *SA* and *IS* denote. (Possibly the label *IS* should also be modified – see the related nomenclature remark above.)

- Would it be possible to indicate celerity values related to the earthquake epicenter and origin time in the horizontal axis?

- What apparent velocity is applied in the generation of the beams plotted in the lowermost station panels?

- For the topmost station data analysis panels, the color scale goes from 0° to 180°, indicating that the display shows the absolute value of the backazimuth deviation? Intuitively, I would assume that plotting the deviation including its sign could provide additional information to the reader, for example regarding the presence of an *azimuthal sweep* as discussed in Section 3?

8) Line 189 and the rest of this paragraph:

- When stating *all four arrays*, I assume you are referring to the four arrays displayed in Figure 2?

- I cannot clearly distinguish the mentioned *quasi-continuous seismic waves* in the IS40 and IS30 displays.

- The reference to *source: USGS* needs to be specified. You could also consider marking the USGS bulletin aftershock arrival times in Figure 2.

- Last sentence: not only the apparent velocity, but also the backazimuth deviation can provide information valuable for identification of these phases. You also could consider moving the last sentence to the beginning of the paragraph.

9) Line 211:

The reported *azimuthal sweep* between 3° and 8° is difficult for me to distinguish

in Figure 2. I would recommend a separate figure which plots backazimuth as a function of time for the different stations during the relevant time window. This would allow for an assessment of the mentioned azimuthal sweep present at IS07 (as well as the absence of a sweep at the other stations).

10) Line 228:

*. . . their later arrival time and lack of high-frequency content correspond to the long lasting signal families following the main signal peak for many minutes in the low frequencies. These signal families can be observed together with low-frequency seismic wave activity and low frequency acoustic components from the stratospheric ducting, discernible only to a certain degree by the apparent velocities and arrival times.*

Is there a way to guide the reader on how to find these thermospheric arrivals in the data shown in Figure 2? For example, you could highlight the relevant time and frequency regimes with boxes.

The second sentence could benefit from being split.

11) Line 270:

I recommend making it more clear to the reader what kind of estimates that were made using the INFERNO software.

Is the claim that the acoustic energy is concentrated around 0.2 Hz based on estimates made in the current study, or is it based on general knowledge from previous works on earthquake-generated infrasound?

Does this refer to the energy spectral content at the source or at the stations?

As a suggestion, you could provide a display of the infrasonic spectral signature(s) at the station(s), and/or an estimate of the spectral components at the source – depending on what is the most relevant.

12) Line 315:

Would it be useful to also list the celerity values found in the simulations?

13) Line 320:

*. . . celerities of those stratospheric ducts to be in the order of 290 m/s.*

Can you quantify what is meant by *in the order of*?

14) Line 329:

My understanding is that the PAPE code supports a range-dependent atmospheric model when simulating the wave propagation and estimating the associated transmission loss. However, from the last sentence in the Figure 4 caption, I get the impression that you indeed consider a 1-D $v_{\text{eff}}$ profile?

If a 1-D approximation of the atmosphere is applied (which might well be an appropriate approximation), I think it should be clarified why this is appropriate, e.g., by showing how much the $v_{\text{eff}}$ profiles vary along the great circle connecting the event and the stations.

15) Line 332:

Maybe you could also show propagation results generated for a lower frequency?

16) Paragraph starting on line 361:

- For what time intervals (and frequency ranges) are the respective station array processing output back-projected?

- The sentence starting with *The uncertainties of the measurements ...* is difficult for me to interpret.

17) Line 377:
The colorbar title *Number of events* can be confusing, because we consider signals from a single earthquake event. Is it appropriate to instead write *Number of PMCC pixels*?

18) Line 393:
As mentioned also above [*Recommendation on specific aspects*, remark 2)], the discussion regarding association between station-data segments and signatures in the consolidated back-projection map would be facilitated if separate maps were provided per station.

19) Line 483:
The relevance of this study to CTBT verification and related infrasound monitoring is not so clear from the text. Maybe links can be provided to the objectives of studies like Assink et al. (2018, 2016); Bowman (2019); Gaebler et al. (2019)? [Which could be given in the Introduction if preferred.]

**Technical remarks and corrections**

1) Title date format:
Looking at previous NHESS papers, it seems like dates in the title are given in the format "28 September 2018", instead of the format "September 28th 2018" used in the submitted manuscript.

2) Line 55:
Is there a reference or DOI available which allows for citing the USGS analysis of the event?

3) Line 65:
Suggested clarification: *The intense ground shaking of either the epicentral region or the or the nearby topography from the Sulawesi earthquake* ⟶ *The intense ground shaking of either the epicentral region or the topography nearby the Sulawesi earthquake*

4) Line 69:
Suggested clarification: *to highly sensitive infrasound arrays* ⟶ *to be recorded at highly sensitive infrasound arrays*

5) Line 83:
Suggested replacement: *Although there is quite a large number of studies* ⟶ *Although there are many studies*

6) Line 88:
Suggested replacement: *of a* ⟶ *related to a*

7) Line 89:
Suggested replacement: *Therefore, one of the main tasks of* ⟶ *Therefore, a main objective*

8) Line 99:
Suggested replacement: *Data from various infrasound arrays of the International Monitoring System (IMS) established under the Comprehensive Nuclear-Test-Ban Treaty (CTBT), are used within this study* $\longrightarrow$
*This study mainly considers data recorded at infrasound arrays of the International Monitoring System (IMS) established under the Comprehensive Nuclear-Test-Ban Treaty (CTBT).*

In addition, a reference could be given to key literature describing the IMS.

9) Line 111:
I think the epicenter symbol shown is rather a star than an asterisk? (Not sure.)

10) Line 197:
Suggested clarification: *since the local infrasound observations generated from* $\longrightarrow$ *microbarometer output generated from*

11) Line 198:
Suggested clarification: *fairly well as seismic arrays here* $\longrightarrow$ *fairly well as seismic arrays for this event*

12) Line 206:
Suggested clarification: *. . . since the back-azimuth calculations as well as the beam-forming are focused on the respective theoretical back-azimuth for the epicenter calculated for each station.* $\longrightarrow$ *. . . where the array beams are focused towards the earthquake epicenter.*

13) Line 237:
As a service to readers not familiar with microbaroms, I would suggest providing a sentence or two plus some key literature reference that explains what is a microbarom signal.

14) Line 243:
Suggested clarification: *acoustic velocities* $\longrightarrow$ *acoustic apparent velocities.*

15) Line 248 (and the final paragraph of Section 3):
To facilitate reading this discussion, islands mentioned in the text could be labeled in at least one of the map figures.

16) Line 255:
This sentence can be hard to follow, I suggest to re-formulate and/or split it.

17) Line 287:
This sentence can be hard to follow, I suggest to re-formulate and/or split it.

18) Line 337:
*The stability of the ducting conditions are best expressed by quantifying the effective sound speed . . .*

I don't expect the effective sound speed ratio is providing the *best* estimate of the acoustic duct availability. For example, wave-propagation modelling can provide a more detailed analysis. Possible alternative formulation:

*The availability of atmospheric ducts can be quantified using the effective sound speed . . .*

19) Line 346:
   This sentence is not so clear to me.

20) Line 396:
   Suggested replacement: *ground movement* $\longrightarrow$ *ground motion.*

21) Line 461:
   I recommend to split and re-formulate the sentences in this paragraph.

22) Line 467:
   I recommend to split and re-formulate the sentences in this paragraph.

23) Line 473:
   This statement is vague – maybe it can be omitted without loss?

24) Line 478, sentence starting with *Taking into account ...*:
   Can this be re-formulated and split?

**References**

J. Assink, G. Averbuch, S. Shani-Kadmiel, P. Smets, and L. Evers. A seismo-acoustic analysis of the 2017 North Korean nuclear test. *Seismological Research Letters*, 89(6): 2025–2033, 2018.

J. D. Assink, G. Averbuch, P. S. M. Smets, and L. G. Evers. On the infrasound detected from the 2013 and 2016 dprk's underground nuclear tests. *Geophysical Research Letters*, 43(7):3526–3533, 2016.

D. C. Bowman. Yield and emplacement depth effects on acoustic signals from buried explosions in hard rock. *Bulletin of the Seismological Society of America*, 109(3):944–958, 2019.

P. Gaebler, L. Ceranna, N. Nooshiri, A. Barth, S. Cesca, M. Frei, I. Grünberg, G. Hartmann, K. Koch, C. Pilger, et al. A multi-technology analysis of the 2017 north korean nuclear test. *Solid Earth*, 10(1):59–78, 2019.

A. Le Pichon, L. Ceranna, and J. Vergoz. Incorporating numerical modeling into estimates of the detection capability of the ims infrasound network. *Journal of Geophysical Research: Atmospheres*, 117(D5), 2012.

K. T. Walker, A. L. Pichon, T. S. Kim, C. de Groot-Hedlin, I.-Y. Che, and M. Garcés. An analysis of ground shaking and transmission loss from infrasound generated by the 2011 tohoku earthquake. *Journal Of Geophysical Research: Atmospheres*, 118(23):12–831, 2013.

J. M. Young and G. E. Greene. Anomalous infrasound generated by the alaskan earthquake of 28 march 1964. *The Journal of the Acoustical Society of America*, 71(2):334–339, 1982.

---

## Referee Comment (RC2) · Alex Iezzi (Referee) · 23 Jul 2019

**Reviewer Comments**

Title: Infrasound and seismoacoustic signatures of the September 28th 2018 Sulawesi super shear earthquake
Authors: Christoph Pilger, Peter Gaebler, Lars Ceranna, Alexis Le Pichon, Julien Vergoz, Anna Perttu, Dorianne Tailpied, Benoit Taisne
MS No.: nhess-2019-179

**General Comments:**

The manuscript entitled "Infrasound and seismoacoustic signatures of the September 28th 2018 Sulawesi super shear earthquake" by Pilger et al explores observations and modeling of both infrasound and seismic signals recorded on select IMS arrays. They incorporate a variety of techniques including array processing, atmospheric propagation modeling, and back projection as a means to understand each of the observations seen on the arrays.

Although they conclude that the observations are not dependent on the fact that the earthquake was super shear, the observations complement the growing field of research on infrasound generated by earthquakes and its contents are of interest to the infrasound community. Therefore, I recommend the paper for publication after moderate revisions and look forward to reading the revised manuscript.

Thank you.

**Specific Comments:**

Line 28:
Do you use time-dependent attenuation and/or propagation modeling in this paper? Or are you referring to using atmospheric conditions close to the earthquake origin time since atmospheric conditions can change over short time scales? If not, please omit this phrase. If so, please add text to the manuscript describing your time-dependent propagation modeling.

Lines 49-51:
Can you provide a reference for this statement?

Line 52:
It may be beneficial to add an inset to Figure 1 with a zoom in on the source region that shows the city and rupture zone in more detail that can be pointed to in the introduction.

Line 79 (and in general):
There seems to be a variety of terms to describe different sources of infrasound (e.g. epicentral, seismoacoustic waves, secondary, etc). I suggest either condensing your definitions to distinguish between epicentral infrasound and secondary infrasound, or explicitly stating how the terms differ in the manuscript.

Lines 100-101:
This is more of a personal preference that you don't have to follow, but it may help highlight your results if instead of starting sentences off with "Figure XX shows", sentences start by stating what the figure shows as the subject of the sentence. For example,
"Figure 1 shows the earthquake epicenter as well as the nearest stations around the event."
Could be changed to
"The earthquake epicenter, as well as the nearest infrasound stations around the event, is shown in Figure 1"

Line 101:
It could be beneficial to give a quick mention of the ranges here, as done in Line 23.

Line 102:
You can mention that the detection/no detection you describe in this paragraph will be shown in the analysis of section 3.

Lines 105-106:
Did you check other IMS arrays? If not, it may be best to refrain from making this assertion.

Line 107:
I know SING is only a single sensor and therefore array processing cannot be performed, but I would suggest adding a figure of the waveform somewhere since it is mentioned it a few times. This can be done as part of figure 2, on its own in the main text, or in supplemental material.

Line 119:
This may be a good place to define apparent velocity and state why it is useful (indicates arrival inclination which can be used to infer propagation path, i.e. higher trace velocity likely indicates arrivals from higher altitude ducts).

122-123:
It could be good to reference Matoza et al (2017) here.
Matoza, R. S., D. N. Green, A. Le Pichon, P. M. Shearer, D. Fee, P. Mialle, andL. Ceranna (2017), Automated detection and cataloging of global explosive volcanism using the International Monitoring System infrasound network, J. Geophys. Res. Solid Earth, 122, 2946–2971, doi:10.1002/2016JB013356.

Line 124:
I suggest being more specific and stating propagation path. Also, "identified" might be a strong word especially at such long distances with uncertainties in the windfiles, possibly "inferred" or "plausible" would be better words? It may also be useful to describe celerity here, as I would argue it is usually a much more decisive propagation path indicator than trace velocity or frequency content (when the origin time of the event is known).

Line 128:

I suggest qualifying this sentence to station that the attenuation map is that of the surface (where the arrays are). This is important because you map stratospheric wind vectors on the same plot and not all readers may infer this.

Lines 144-147:
For merging the lower atmospheric ECMWF files with upper atmospheric climatologies, did you use Doug Drob's G2S model? If so, this should be explicitly stated and Drob et al (2003) should be cited.

Lines 148-158:
I believe this methodology does not account for atmospheric variability (which caused location biases in the Shani-Kadmiel et al (2017) paper). Please specify either way in the manuscript here or in the discussion section.

Lines 161 – 165:
Please state your filter bands used in the manuscript text (it is only stated in the figure caption)

Line 165:
It may be good to mention back-azimuth here as that is the term that is used in the rest of the paper
"direction of origin (back-azimuth)"

Lines 166:
I assume you mean 4 infrasound arrays that made detections?

Line 168:
Celerity should be defined earlier (see comment for line124). Also, a more precise definition of celerity would be "the horizontal distance between source and receiver divided by the total traveltime".

Table 1:
The expected arrival time row may not be necessary, as it requires the assumption of a stratospheric celerity (0.3 km/s), which is not always the correct propagation path.
Also, please state in the text how is signal duration defined.

Figure 2:
Adding subplot labels (a, b, c) that can be referred to in the results section may help support your claims and help the reader follow along.
The font sizes might be slightly small in this figure.

Line 188:
The list format of your results section is a bit odd. I suggest removing this line and making the bullets into paragraphs.

Line 202:

It might be nice to add a plot of the winds as supplemental material (even if it is just one sonde above the source) and discuss if the back azimuth deviations are consistent with the cross winds of the specified propagation duct.

Line 241:
This sentence is unnecessary and could be removed.

Line 243:
Please clarify the phrase "acoustic velocities". Do you mean apparent velocity?

Line 244:
It should be qualified that celerities outside these bounds exclude purely acoustic waves from the origin at the time of rupture.

Lines 260-265:
Please be more clear on your definitions of both attenuation and propagation modeling. You calculate them using different methods, but they both show results of transmission loss with your attenuation shown in map view and the propagation modeling shown as a cross section.

Line 268:
What is the spatial resolution of the range dependent atmospheric profiles? 0.5 x 0.5 degree as stated in Figure 3 caption? Please state this in the manuscript text.

Line 270:
Please elaborate on what INFERNO is and how you use it.

Lines 266-295:
The transmission losses, associated uncertainties for each array, and source frequency in this paragraph could all be put into a table to more concise and easily get your point across to the reader.

Figure 4:
Why did you choose 1Hz modeling? That seems a bit high for propagation of such large distances.

Figure 5:
Lines 384-395 refer to Sulawesi Island, North Maluku, etc. It might be helpful to label these areas in the figure so the reader can more easily follow along. Or, if you chose to add an inset to Figure 1 you can refer to the locations there.

Line 425:
Tiny semantic issue, but I don't believe the earthquake is named after the mountain. The earthquake ruptured the Denali fault as well as the Toschunda fault and is located in/near Denali National Park.

Line 449:

Do you have a reference for this?

Line 463:
I would argue that both topography and conducive propagation conditions are necessary for detection.

**Technical Corrections:**

Line 25
"is supposed to" sounds a bit awkward
Perhaps "The seismic-to-acoustic coupling at nearby terrain features *is shown to* generate distinct infrasonic signatures clearly recordable at remote infrasound arrays.

Line 27:
Suggest adding the word "infrasound" for clarity.
Event-related *infrasound* observations

Lines 30, 95, 148, 358:
In these lines, "back tracking" is used. Please choose one term (I think "back projection" is the most widely used) and use throughout the manuscript.

Line 41:
Do you mean "a very high *rate of* natural seismicity"?

Line 63:
"in the course of" may better be stated as "surrounding this event"

Line 65:
Would it be more appropriate to say "both" instead of "either"?

Line 179:
I don't think "from above" is necessary.

Line 196:
There should be a better way to cite this. Please check the NHESS citation guidelines.

Figure 3:
The stratospheric wind vectors pretty hard to see. Can you make them slightly larger?

---

## Author Comment (AC1) · 2 Sep 2019

*Dear Peter Näsholm,*

*thank you very much for taking the time and interest to deliver this very thorough and constructive review of our manuscript. We carefully studied your comments and made changes and corrections to the manuscript where necessary. We hope our changes and corrections are sufficient to make our article suitable for publication soon. Your comments and suggestions certainly helped to improve quality and clarity of the paper.*

*Thank you again, best regards*

*Christoph Pilger and co-authors*

**RESPONSES to Reviewer 1** (provided below each reviewer comment in green):

Recommendations on specific aspects

1) Super-shear related analyses:

I recommend providing a more concise background on why it is interesting to understand if the observed strong infrasound waves are related to the super-shear nature of the earthquake. Then the discussion and comparison with other earthquakes can be linked to the added background information. For example, it could be explained more precisely what is a super-shear earthquake and it could be mentioned that these can be associated with an effect analogous to acoustic sonic booms with the source traveling faster than the wave propagation speed.

Since the detectability is naturally influenced by the presence of infrasound arrays within the waveguides available from the source, the authors could consider complementing Table 2 with additional information on the infrasound array network available around each of the listed events.
What are the criteria used for the assessments *weak infrasound* and *strong infrasound* in Table 2? I would also recommend stating the distance between event and station, instead of no or vague information about distance (*nearby / remote*).

It would be informative with a concise explanation why the registrations (or non-registrations) of seismic arrivals on the infrasound stations are relevant to the assessment? (Otherwise, I would recommend removing this information from Table 2)

RESPONSE: We added a more concise description and background information to super-shear earthquakes to the second paragraph of the introduction (including comparison to sonic booms, shear wave mach cone, increased ground motion and references).
We modified table 2 (now table 3) to not just mention selected IMS stations detecting the event, but to mention all IMS stations available (and data-providing) at the given time within a radius of 5000 km. We separate them between "event detection" and "no event detection". This should highlight how the detectability is related to the available stations, also providing their distances to the respective event epicenter.
We added the signal duration to each detection, but we removed the assessments of "weak" and "strong" infrasound in the table, since there's no objective criterion to derive that. The amplitude naturally decreases with distance, whereas amplitude and thus also signal-to-noise ratio depends on the chosen filter-bands as well.
The distances qualifications of "nearby" and "remote" were removed in favor of precise distances per station. We removed the information of seismic arrivals from the table following the reviewer's suggestion.

2) Back-projection analysis:

I would recommend reinforcing the discussion on the back-projection analysis. Based on the current displays, I find it hard to analyze what are the key wavetrain segments on the different stations that contribute to the different highlighted geographical regions shown in Fig. 5.

Maybe it would be useful to provide an additional back-projection result map per station, either in the paper body or in an appendix? This might make it more straightforward to the reader to see, e.g., if the two blue colored source regions at around 127°E are related to signals on one or on multiple stations. This could also help providing insight in the way each of the stations constrains the source region estimates on Sulawesi. As a suggestion, the coloring of the added per-station maps might be a function of the reduced time for the corresponding infrasound detections? This could then allow for an interpretation of links between array parameter output as a function of time (and the associated waveforms) in the context of the different source regions and features.

A separate back-projection image per station would also allow the reader to better follow and assess the discussion in the last paragraph of Section 3.

RESPONSE: We added additional back projection maps for each station in the supplementary section. They support the discussion in the text, e.g. pointing out that the 127°E sources are just in the I40PG data (and possibly related to something else as already stated in the text). The source constraints for each station should be clearer identified and the major influence of I07AU on the PMCC pixel maxima should be seen as well. It is hardly possible to generate a color-coded event density map on the one hand and provide coloring for the timing on the other hand, so (for supplementary material) we keep with the event density coloring. The supplementary figures are mentioned in the manuscript text at the first introduction of the cumulative figure 5 and at multiple places during the discussing to allow the reader to better follow and assess it, as suggested by the reviewer.

3) Suggested references to add:

• Studies including back-projection of infrasound recordings: Assink et al. (2018); Walker et al. (2013).
• A study which shows observations of infrasonic waves generated by seismic surface waves along the Rocky Mountains: Young and Greene (1982).

RESPONSE: References were added to the introduction and the discussion, where they fitted best.

4) Atmospheric model product:

From which ECMWF atmospheric model product are the wind and temperature fields extracted?

RESPONSE: The product was specified in the data section: "Atmospheric wind and temperature profiles are derived from operational high-resolution atmospheric model analysis, defined by the Integrated Forecast System of the ECMWF, available at https://www.ecmwf.int/ ".

5) Transmission loss:

The transmission loss is estimated both using parabolic equation simulations in a lossy medium, and using a semi-empirical formula. However, I find a potential confusion in the mixed use of the concepts of attenuation and transmission loss in the text. Figure 3 and Figure 4 display the results of transmission loss estimates using the two techniques. If you agree on this viewpoint, I propose to streamline the language and the discussion of the associated results to underline that both approaches are applied to calculate transmission loss.

For example, line 137 can be confusing. It could be modified from:
*In both attenuation and propagation modeling, data from the European . . .*
into:
*In both the semi-empirical and the parabolic equation-based transmission loss estimates, data from the European . . . .*

Moreover, it would be interesting to see a direct comparison between the two trans- mission loss estimates as calculated for the station sites.

RESPONSE: We streamlined the manuscript to replace the mixed language concept of attenuation and transmission loss modeling by the sole use of the term transmission loss. We keep the concept of propagation modeling as a second technique and clarified the differences and benefits of those two concepts both applied in the study: "Transmission loss calculations using firstly a semi-empirical method for a horizontal representation (map view, figure 3) and secondly a parabolic-equation-based propagation model for a vertical representation (cross section, figure 4) are performed...". We modified line 137 according to the reviewer's suggestion and screened the manuscript for further passages to be changed in this context.

6) Nomenclature:

• I suggest homogenizing the wording related to the concept of *back-projection*. The paper also uses *backtracking* and *back-tracking*, while some other works in addition consider a more general concept of *back-propagation*. If the mix is intentional, a more concise explanation is needed. Otherwise, I suggest using only one of the constructs in the text.

• The paper could be clearer and more consequent on the use of the different physical infrasound and seismic wave generation processes considered in observations, analyses, and figure labels. Currently, the constructs *epicentral infrasound; seismoacoustic signals; secondary infrasound; seismoacoustic precursors; seismoacoustic successors; secondary signals* are used, and I think the readers would appreciate a consolidated use and definition of these concepts.
I'm not sure whether the definition of *seismoacoustic waves* provided on Line 79 is appropriate:
*. . . also highlight infrasound generated from secondary phenomena like remote ground motion of mountain chains or extended basin areas, and from tsunami waves hitting the coastline. This secondary infrasound is often called seismoacoustic waves, . . .*
Is epicentral infrasound supposed to be included in your definition of a seismoa- coustic waves?

RESPONSE:
- The nomenclature was modified to a homogenized use of "back projection" throughout the manuscript according to both reviewers' suggestions
- The nomenclature was harmonized regarding the use of epicentral and seismoacoustic sources. We harmonized the manuscript to no further use the wordings "secondary" (after the introductory definition), as well as no further use of "precursors" and "successors" and keep with only the two concepts of "epicentral infrasound" and "seismoacoustic".
- Infrasound from epicentral ground movement is defined in paragraph 3 of the introduction, seismoacoustic waves by remote terrain features in paragraph 4 of the introduction. I think the sentence clarified that epicentral infrasound is NOT included in the definition of seismoacoustics (which is described needing "seismic waves" and "distant terrain features"). To reduce confusion, we moved the mentioning of tsunami waves away from the end of the sentence and added a clarifying repetition of "by remote ground motion".

**Specific text edit suggestions**

1) Line 107:

SING data is not shown, which makes it difficult to assess the summary given on these data signatures. I propose including a SING signal plot either in the paper body or in an appendix.

RESPONSE: A SING signal plot and an INFERNO analysis of it is added to the supplement, both giving a representation of the SING station data and the INFERNO method. Manuscript text refers to the supplementary figure in two adequate passages.

2) Line 116:

I would advise including a reference or explanation to describe what is an NDC and what is the NDC-in-a-box software.

RESPONSE: There is no reference, at least in paper, about the NDC-in-a-box. Nevertheless we added some lines describing the NDC-in-a-box software as well as the PMCC application in more detail. We also added an extended description of NDC, since it is the first appearance of this acronym.

3) Line 124:

It needs to be better clarified what is meant when claiming that *the propagation of signals* can be identified from the apparent velocity and frequency content.

I assume this would be related to identification of the atmospheric ducts penetrated by the acoustic waves (tropospheric / stratospheric / mesospheric / lower thermospheric?

RESPONSE: the "propagation" was changed to "propagation path" and stratospheric and thermospheric ducting was added for clarification, including a reference to Drob et al. 2003, following the remark of both reviewers. Since this is only the method section, we do not explain how the method-derived apparent velocity and frequency content are used for discriminating seismic and acoustic signals as well as stratospheric or thermospheric ducted infrasounds. This is done later on in section 3.

4) Line 130:

I suggest including a concise statement of the parameters that go into the calculation, preferably complemented by the full equation for the semi-empirical transmission loss estimates. Is this equation 2 in Le Pichon et al. (2012)?

RESPONSE: The method is the same as used in equation 2 of Le Pichon et al., 2012, later enhanced in Tailpied et al., 2017. The method is already described in the text to be frequency- and range-dependent and depending on propagation calculations, all around the referred line 130. Later on, in the next paragraph, the effective sound speed (ratio) is described as the one most important parameter used for both methods. We think that it is unnecessary and probably misleading to add, explain and discuss the full formula here or to give more space for the method description, since it is referenced by two actual papers both containing and describing the method and the formula in larger detail. A third paper is in preparation but cannot be referenced before submission and acceptance.

5) Line 165:

How is the signal duration estimated?

RESPONSE: Signal duration estimation is described at the first text passage in section 3 where actual durations of 25 and 44 min for I39PW and I07AU are discussed: it is "derived from the width of the high-frequency part signals originating from epicentral directions in the PMCC analyses".

6) Line 168:
I'm not sure celerity is best described as *speed over ground*? Maybe you could use, e.g., *ratio of the range to the traveltime*, or *epicentral distance divided by the traveltime*?

RESPONSE: we modified the description of celerity according to the reviewer's suggestion.

7) Line 178 (Figure 2):

• Would a separation of this figure into four labeled subfigures (one per station) make the grouping of the subpanels more apparent? I think the readability of Figures 2–4 would also benefit from using a larger text label font (or larger figures).

• The middle panel labels for each station say *signal speed*, while the paper text otherwise uses *apparent velocity*. I think this should be consolidated.

• Suggested clarification: *are sorted by distance from above* ↦ *are ordered by epicentral distance*.

• The caption should explain what the labels *SA* and *IS* denote. (Possibly the label *IS* should also be modified – see the related nomenclature remark above.)

• Would it be possible to indicate celerity values related to the earthquake epicenter and origin time in the horizontal axis?

• What apparent velocity is applied in the generation of the beams plotted in the lowermost station panels?

• For the topmost station data analysis panels, the color scale goes from $0°$ to $180°$, indicating that the display shows the absolute value of the backazimuth deviation? Intuitively, I would assume that plotting the deviation including its sign could provide additional information to the reader, for example regarding the presence of an *azimuthal sweep* as discussed in Section 3?

RESPONSE:
- We labeled the subfigures with a,b,c,d (one per station) for a more apparent grouping of the subplots. Label text size was slightly increased, but otherwise much larger text fonts were not possible since too much information would be hidden or suppressed when enlarging the fonts. Discussion with the journal can specify in the end if the final size of the figure allows text readability or if further modifications are necessary.
- Signal speed is changed to apparent velocity in the figure in a consolidated way
- Clarifications were made to the caption regarding figure order and "IS"/"SA" labels
- Regarding celerity information in the figure 2 as a horizontal axis (similar as to the font size reply): more information added to the already quite dense figure 2 is not possible due to available space and would in our opinion not help the reader much more. We refer to table 1 for celerity values
- Apparent velocities used for the beamforming: used are the observed mean back-azimuth and apparent velocity values found for the whole PMCC pixel family, as listed in the last two rows of table 1
- We added supplementary figures which do not focus on absolute deviation of back-azimuth but on a narrow range of back-azimuths. This highlights the azimuthal sweep at I07AU and includes additional information, e.g. a larger frequency distribution at I39PW. See separate comment 9 for further details.

8) Line 189 and the rest of this paragraph:

- When stating *all four arrays*, I assume you are referring to the four arrays displayed in Figure 2?

- I cannot clearly distinguish the mentioned *quasi-continuous seismic waves* in the IS40 and IS30 displays.

- The reference to *source: USGS* needs to be specified. You could also consider marking the USGS bulletin aftershock arrival times in Figure 2.

- Last sentence: not only the apparent velocity, but also the backazimuth deviation can provide information valuable for identification of these phases. You also could consider moving the last sentence to the beginning of the paragraph.

RESPONSE:
- yes, all four arrays displayed in Figure 2 were meant, clarification added to the manuscript (also considering next bullet point changes)
- everything with apparent velocities above 1 km/s (blue) and azimuth directions towards the epicenter (yellow) in the low frequencies (0.05Hz – 0.5Hz) following the event is mentioned to be "quasi-continuous" seismic waves. They last for about 1-2 hours following the event, in IS40 it is quite similar to IS07 and IS39. Maybe it is difficult to say "continuous", this is why it was stated "quasi", and especially I30JP data, after reconsideration, is mixed up with aftershocks or other seismic activity near Japan and from different directions. Maybe the distance to I30JP is also too large to see seismic-wave-shaking of the instruments. Taking this into account the manuscript text about I30JP was modified to state "and possibly I30JP".
- Reference is specified to the USGS event page for the Sulawesi 2018 event, also see other comments by both reviewers to this. Since the seismic waves are not the focus of this study, we chose to not give any markers to the main seismic event or aftershocks in the figure, it seems the figure is crowded enough.
- backazimuth information is added to the sentence. The sentence could be in the beginning of the paragraph, yes, kind of introducing seismic waves at infrasound sensors, but it is also positioned not too bad for a conclusion that infrasound sensors work fairly well as seimic sensors. In our opinion it is better stated here, since the seismic observations are then already described, and we keep it as is.

9) Line 211:
The reported *azimuthal sweep* between 3° and 8° is difficult for me to distinguish in Figure 2. I would recommend a separate figure which plots backazimuth as a function of time for the different stations during the relevant time window. This would allow for an assessment of the mentioned azimuthal sweep present at IS07 (as well as the absence of a sweep at the other stations).

RESPONSE: A supplementary figure was added which highlights the azimuthal sweep at I07AU and certain other features as well as the absence of such an explicit sweep at the other stations. It is referenced in the manuscript text where the azimuthal sweep is mentioned. The according lines were modified to provide the necessary information about azimuth deviations pointing at the supplementary figure and it's caption for further details.

10) Line 228:
*...their later arrival time and lack of high-frequency content correspond to the long lasting signal families following the main signal peak for many minutes in the low frequencies. These signal families can be observed together with low-frequency seismic*

*wave activity and low frequency acoustic components from the stratospheric ducting, discernible only to a certain degree by the apparent velocities and arrival times.*

Is there a way to guide the reader on how to find these thermospheric arrivals in the data shown in Figure 2? For example, you could highlight the relevant time and frequency regimes with boxes.

The second sentence could benefit from being split.

RESPONSE: The manuscript text was modified to name the approximate time and frequency regimes of the thermospheric arrivals. Similar to other replies, we think that figure 2 is already crowded enough and introducing further axes, boxes and areas do not support the reader in mastering this figure. We split the sentence in two parts according to the reviewer's suggestion.

11) Line 270:

I recommend making it more clear to the reader what kind of estimates that were made using the INFERNO software.

Is the claim that the acoustic energy is concentrated around 0.2 Hz based on estimates made in the current study, or is it based on general knowledge from previous works on earthquake-generated infrasound?

Does this refer to the energy spectral content at the source or at the stations?

As a suggestion, you could provide a display of the infrasonic spectral signature(s) at the station(s), and/or an estimate of the spectral components at the source – depending on what is the most relevant.

RESPONSE: The INFERNO software, its key features and its application within this study is elaborated in more detail in the given passage, following the recommendation of both reviewers. The energy concentration around 0.2 Hz was derived by applying INFERNO on station data for the current study. A display of the single sensor SING spectral components using INFERNO is provided as a supplementary figure.

12) Line 315:

Would it be useful to also list the celerity values found in the simulations?

RESPONSE: Celerity values were derived from ray-tracing simulations (see next comment: 287-293 m/s), but it is more difficult to derive celerity values from a PE, since the output is a 2-D transmission loss field and not a path that you can follow with a certain travel time. So we stick to quantifying celerity by the raytracing and the observations.

13) Line 320:

*... celerities of those stratospheric ducts to be in the order of 290 m/s.*

Can you quantify what is meant by *in the order of*?

RESPONSE: Text passage specified to the model values between 287 and 293 m/s.

14) Line 329:

My understanding is that the PAPE code supports a range-dependent atmospheric model when simulating the wave propagation and estimating the associated transmission loss. However, from the last sentence in the Figure 4 caption, I get the impression that you indeed consider a 1-D $v_{eff}$ profile?

If a 1-D approximation of the atmosphere is applied (which might well be an appropriate approximation), I think it should be clarified why this is appropriate, e.g., by showing how much the $v_{eff}$ profiles vary along the great circle connecting the event and the stations.

RESPONSE: Range-dependent PE modeling is performed (so 2-dimensional veff profiles), the last sentence of the figure 4 caption was indeed quite misleading. We changed it accordingly: "An averaged effective sound speed profiles (veff) is shown for each station.". 1-D approximation is not applied.

15) Line 332:
   Maybe you could also show propagation results generated for a lower frequency?

RESPONSE: The choice of 1 Hz is a compromise between the 0.2 Hz (figure 3a) of the low-frequency acoustic energy maximum and the 3 Hz (figure 3b) of the high-frequency signal content which is unique for the stratospheric ducted signal parts. It is also a matter of necessity to be able to present the stratospheric ducts in a clear and concise figure. When going to lower frequencies, there is less attenuation and higher wavelengths and the modeling as well as picture becomes less detailed and more blurred, losing the clear optical representation of the stratospheric waveguides which are the main message of the figure.

16) Paragraph starting on line 361:
   • For what time intervals (and frequency ranges) are the respective station array processing output back-projected?

   • The sentence starting with *The uncertainties of the measurements . . .* is difficult for me to interpret.

RESPONSE:
- The complete station array output shown in figure 2 is back-projected, also covering the complete frequency range shown from 0.01 to 4.4 Hz. The only main restriction is that the signals for PMCC internal back projection are limited to a back-azimuth angle pointing towards the epicenter plus/minus 40 degree (we tried versions with 10, 20 and 40 degree and chose the largest version to keep any signal potentially associated to the earthquake). Of course this might exclude some (singular) pixels from directions larger than 40 degree back azimuth deviation, but those might be unassociated anyway. In this case the majorities of signals by microbaroms and surf are excluded by this approach and epicentral or seismoacoustic signals potentially related to the earthquake are included. We added this previously unmentioned +/- 40 degree maximum deviation to the manuscript text.
- We rephrased the sentence about the uncertainties for this method and split it into three parts for clarification.

17) Line 377:
   The colorbar title *Number of events* can be confusing, because we consider signals from a single earthquake event. Is it appropriate to instead write *Number of PMCC pixels*?

RESPONSE: the title "Number of events is changed to "Number of PMCC pixels" according to the reviewer's suggestion.

18) Line 393:
   As mentioned also above [*Recommendation on specific aspects*, remark 2)], the discussion regarding association between station-data segments and signatures in the consolidated back-projection map would be facilitated if separate maps were provided per station.

RESPONSE: Separate maps are provided in the supplementary section, see authors reply to the Recommendation on specific aspects, remark 2, for further details.

19) Line 483:
   The relevance of this study to CTBT verification and related infrasound monitoring is not so clear from the text. Maybe links can be provided to the objectives of studies like Assink et al. (2018, 2016); Bowman (2019); Gaebler et al. (2019)? [Which could be given in the Introduction if preferred.]

RESPONSE: Although we find it unusual to end a publication with references, we follow the reviewer's suggestion (it did not fit so well in the introduction either) and add a link and the references to other studies.

**Technical remarks and corrections**

1) Title date format:
   Looking at previous NHESS papers, it seems like dates in the title are given in the format "28 September 2018", instead of the format "September 28th 2018" used in the submitted manuscript.

RESPONSE: date format modified throughout the manuscript according to reviewers suggestion.

2) Line 55:
   Is there a reference or DOI available which allows for citing the USGS analysis of the event?

RESPONSE: there is a USGS event page to the Sulawesi 2018 earthquake, which we referenced according to the reviewer's suggestion.

3) Line 65:
   Suggested clarification: *The intense ground shaking of either the epicentral region or the or the nearby topography from the Sulawesi earthquake* → *The intense ground shaking of either the epicentral region or the topography nearby the Sulawesi earthquake*

RESPONSE: modified according to reviewer's suggestion.

4) Line 69:
   Suggested clarification: *to highly sensitive infrasound arrays* → *to be recorded at highly sensitive infrasound arrays*

RESPONSE: modified according to reviewer's suggestion.

5) Line 83:

Suggested replacement: *Although there is quite a large number of studies* ⟶ *Although there are many studies*

RESPONSE: modified according to reviewer's suggestion.

6) Line 88:
   Suggested replacement: *of a* ⟶ *related to a*

RESPONSE: modified according to reviewer's suggestion.

7) Line 89:
   Suggested replacement: *Therefore, one of the main tasks of* ⟶ *Therefore, a main objective*

RESPONSE: modified according to reviewer's suggestion.

8) Line 99:
   Suggested replacement: *Data from various infrasound arrays of the International Monitoring System (IMS) established under the Comprehensive Nuclear-Test-Ban Treaty (CTBT), are used within this study* ⟶
   *This study mainly considers data recorded at infrasound arrays of the International Monitoring System (IMS) established under the Comprehensive Nuclear-Test-Ban Treaty (CTBT).*

   In addition, a reference could be given to key literature describing the IMS.

RESPONSE: modified according to reviewer's suggestion. References added to the Le Pichon Springer books for IMS and infrasound array descriptions.

9) Line 111:
   I think the epicenter symbol shown is rather a star than an asterisk? (Not sure.)

RESPONSE: Yes, changed to "star".

10) Line 197:
    Suggested clarification: *since the local infrasound observations generated from* ⟶ *microbarometer output generated from*

RESPONSE: modified according to reviewer's suggestion.

11) Line 198:
    Suggested clarification: *fairly well as seismic arrays here* ⟶ *fairly well as seismic arrays for this event*

RESPONSE: modified according to reviewer's suggestion.

12) Line 206:
    Suggested clarification: *... since the back-azimuth calculations as well as the beam-forming are focused on the respective theoretical back-azimuth for the epicenter*

*calculated for each station.* →  . . . *where the array beams are focused towards the earthquake epicenter.*

RESPONSE: shortened and clarified similar to the reviewer's suggestion.

13) Line 237:
As a service to readers not familiar with microbaroms, I would suggest providing a sentence or two plus some key literature reference that explains what is a microbarom signal.

RESPONSE: a short explanation to what are microbaroms is added to the manuscript as well as two literature references for further reading.

14) Line 243:
Suggested clarification: *acoustic velocities* → *acoustic apparent velocities.*

RESPONSE: modified according to reviewer's suggestion.

15) Line 248 (and the final paragraph of Section 3):
To facilitate reading this discussion, islands mentioned in the text could be labeled in at least one of the map figures.

RESPONSE: Islands are labeled in figure 1 (Borneo and Sulawesi) as well as in figure 5 (Borneo, Sulawesi, North Maluku) according to the reviewer's suggestion.

16) Line 255:
This sentence can be hard to follow, I suggest to re-formulate and/or split it.

RESPONSE: We slightly re-formulated the sentence and split it into two parts for clarification.

17) Line 287:
This sentence can be hard to follow, I suggest to re-formulate and/or split it.

RESPONSE: this sentence and the following one were slightly re-formulated for clarification.

18) Line 337:
*The stability of the ducting conditions are best expressed by quantifying the effective sound speed . . .*

I don't expect the effective sound speed ratio is providing the *best* estimate of the acoustic duct availability. For example, wave-propagation modelling can provide a more detailed analysis. Possible alternative formulation:

*The availability of atmospheric ducts can be quantified using the effective sound speed . . .*

RESPONSE: modified according to reviewer's suggestion.

19) Line 346:
This sentence is not so clear to me.

RESPONSE: The sentence was restated to improve clarification.

20) Line 396:
Suggested replacement: *ground movement* $\longrightarrow$ *ground motion*.

RESPONSE: modified according to reviewer's suggestion.

21) Line 461:
I recommend to split and re-formulate the sentences in this paragraph.

RESPONSE: Sentences were split and re-formulated according to the reviewer's suggestion.

22) Line 467:
I recommend to split and re-formulate the sentences in this paragraph.

RESPONSE: Sentences were split and re-formulated according to the reviewer's suggestion.

23) Line 473:
This statement is vague – maybe it can be omitted without loss?

RESPONSE: statement omitted according to reviewer's suggestion.

24) Line 478, sentence starting with *Taking into account . . .* :
Can this be re-formulated and split?

RESPONSE: Sentence was split and re-formulated according to the reviewer's suggestion.

[revised manuscript text omitted]

---

## Author Comment (AC2) · 2 Sep 2019

*Dear Alex Iezzi,*

*thank you very much for taking the time and interest to deliver this very thorough and constructive review of our manuscript. We carefully studied your comments and made changes and corrections to the manuscript where necessary. We hope our changes and corrections are sufficient to make our article suitable for publication soon. Your comments and suggestions certainly helped to improve quality and clarity of the paper.*

*Thank you again, best regards*

*Christoph Pilger and co-authors*

**RESPONSES to Reviewer 2** (provided below each reviewer comment in green):

Specific Comments:

Line 28:
Do you use time-dependent attenuation and/or propagation modeling in this paper? Or are you referring to using atmospheric conditions close to the earthquake origin time since atmospheric conditions can change over short time scales? If not, please omit this phrase. If so, please add text to the manuscript describing your time-dependent propagation modeling.

RESPONSE: According to the reviewer's suggestion we omit the phrase time-dependent. It is a reference to the atmospheric conditions close to the earthquake origin time, yes, but not a reference to time-dependent modeling, which we do not perform.

Lines 49-51:
Can you provide a reference for this statement?

RESPONSE: The reference would be the USGS archive of earthquakes related to the Sulawesi one, as stated in the data availability section in the end of the paper. We rephrased the sentence to mention the USGS Sulawesi earthquake event page as a literature reference, also quantifying the number of related events described in these lines.

Line 52:
It may be beneficial to add an inset to Figure 1 with a zoom in on the source region that shows the city and rupture zone in more detail that can be pointed to in the introduction.

RESPONSE: Instead of an inset, a second subfigure was added to figure 1, zooming in on the source region and showing the rupture zone as well as the city of Palu. The figure caption was changed accordingly.

Line 79 (and in general):
There seems to be a variety of terms to describe different sources of infrasound (e.g. epicentral, seismoacoustic waves, secondary, etc). I suggest either condensing your definitions to distinguish between epicentral infrasound and secondary infrasound, or explicitly stating how the terms differ in the manuscript.

RESPONSE: According to the remarks of both reviewers for a harmonized use and definition of the two concepts of infrasound sources studied here, we modified the whole manuscript (see extended comment to reviewer1's similar remark) and keep with either "epicentral infrasound" or "seismoacoustic". We removed the other terms.

Lines 100-101:
This is more of a personal preference that you don't have to follow, but it may help highlight your results if instead of starting sentences off with "Figure XX shows", sentences start by stating what the figure shows as the subject of the sentence. For example,
"Figure 1 shows the earthquake epicenter as well as the nearest stations around the event."
Could be changed to
"The earthquake epicenter, as well as the nearest infrasound stations around the event, is shown in Figure 1"

RESPONSE: We agree with the reviewer's suggestion and changed the phrase accordingly. We also checked and modified the manuscript regarding further phrases like "figure xx shows".

Line 101:
It could be beneficial to give a quick mention of the ranges here, as done in Line 23.

RESPONSE: Distances were added according to the reviewer's suggestion.

Line 102:
You can mention that the detection/no detection you describe in this paragraph will be shown in the analysis of section 3.

RESPONSE: references to section 3 were added according to the reviewer's suggestion.

Lines 105-106:
Did you check other IMS arrays? If not, it may be best to refrain from making this assertion.

RESPONSE: Yes, we checked other more remote IMS arrays for signals from the 28 September 2018 Sulawesi earthquake and found no indications related to this event.

Line 107:
I know SING is only a single sensor and therefore array processing cannot be performed, but I would suggest adding a figure of the waveform somewhere since it is mentioned it a few times. This can be done as part of figure 2, on its own in the main text, or in supplemental material.

RESPONSE: A SING signal plot and an INFERNO analysis of it is added to the supplement, both giving a representation of the SING station data and the INFERNO method mentioned in the manuscript. Manuscript text now refers to the supplementary figure in two adequate passages.

Line 119:
This may be a good place to define apparent velocity and state why it is useful (indicates arrival inclination which can be used to infer propagation path, i.e. higher trace velocity likely indicates arrivals from higher altitude ducts).

RESPONSE: We added explanations to back-azimuth and apparent velocity and their use to derive direction and inclination of signal arrivals.

122-123:
It could be good to reference Matoza et al (2017) here.
Matoza, R. S., D. N. Green, A. Le Pichon, P. M. Shearer, D. Fee, P. Mialle, and L. Ceranna (2017), Automated detection and cataloging of global explosive volcanism using the International Monitoring System infrasound network, J. Geophys. Res. Solid Earth, 122, 2946–2971, doi:10.1002/2016JB013356.

RESPONSE: Agreed. Reference was added to manuscript and literature.

Line 124:
I suggest being more specific and stating propagation path. Also, "identified" might be a strong word especially at such long distances with uncertainties in the windfiles, possibly "inferred" or "plausible" would be better words? It may also be useful to describe celerity here, as I would argue it is usually a much more decisive propagation path indicator than trace velocity or frequency content (when the origin time of the event is known).

RESPONSE: we stated propagation "path" and changed "identified" to "inferred", according to the reviewer's suggestion. We do not describe celerity here, since the paragraph is a description of PMCC derived parameters and the information from them. To use celerity, specific source information is needed to calculate traveltime and distance and derive celerity from it. Celerity will be described in more detail later on according to Reviewer1's Specific comment #6.

Line 128:

I suggest qualifying this sentence to station that the attenuation map is that of the surface (where the arrays are). This is important because you map stratospheric wind vectors on the same plot and not all readers may infer this.

RESPONSE: clarification was made according to the reviewer's comment by adding "at surface level" to the given line.

Lines 144-147:
For merging the lower atmospheric ECMWF files with upper atmospheric climatologies, did you use Doug Drob's G2S model? If so, this should be explicitly stated and Drob et al (2003) should be cited.

RESPONSE: No, Doug Drob's model G2S is not used, but a self-constructed merging approach of the authors of this study.

Lines 148-158:

I believe this methodology does not account for atmospheric variability (which caused location biases in the Shani-Kadmiel et al (2017) paper). Please specify either way in the manuscript here or in the discussion section.

RESPONSE: A statement of the method not accounting for atmospheric variability and thus introducing location biases in addition to the measurement uncertainty and the bias by using a fixed celerity was added to the second paragraph of the discussion. The method nevertheless proves to provide realistic estimates since the averaged conditions used (300 m/s stratospheric ducted signal) are quite near to the ones calculated from observations at the three nearest stations (stated in the text).

Lines 161 – 165:
Please state your filter bands used in the manuscript text (it is only stated in the figure caption)

RESPONSE: we added the filter bands used in the manuscript text.

Line 165:
It may be good to mention back-azimuth here as that is the term that is used in the rest of the paper
"direction of origin (back-azimuth)"

RESPONSE: we modified the phrase according to the reviewer's suggestion. A definition/explanation of back-azimuth was already given in response to a previous comment (related to line 119 above).

Lines 166:
I assume you mean 4 infrasound arrays that made detections?

RESPONSE: Yes, "infrasound" was added in the text.

Line 168:
Celerity should be defined earlier (see comment for line124). Also, a more precise definition of celerity would be "the horizontal distance between source and receiver divided by the total traveltime".

RESPONSE: Corresponding to our statement for line 124, we do not define celerity before and mention it here for the first time. In agreement with reviewer1's specific comment #6, we provide a more precise definition of celerity here following the reviewers' suggestions.

Table 1:
The expected arrival time row may not be necessary, as it requires the assumption of a stratospheric celerity (0.3 km/s), which is not always the correct propagation path.
Also, please state in the text how is signal duration defined.

RESPONSE: expected arrival times may not be correct if the celerity is different from 300. Nevertheless it is by definition ("expected") an assumption and helps the reader and the discussion in the text to compare observations to expectations, so we keep the row. A definition of the use of signal duration ("derived from the width of the high-frequency part signals originating from epicentral directions in the PMCC analyses") is added to the manuscript text.

Figure 2:
Adding subplot labels (a, b, c) that can be referred to in the results section may help support your claims and help the reader follow along.
The font sizes might be slightly small in this figure.

RESPONSE: Subplot labels (a-d) were added to figure 2, according to both reviewers' remarks. Label font sizes were slightly increased, but a larger increase is difficult since the figure is already quite dense. See reply to reviewer 1 for further details.

Line 188:
The list format of your results section is a bit odd. I suggest removing this line and making the bullets into paragraphs.

RESPONSE: This part of the manuscript was modified according to the reviewer's suggestion.

Line 202:

It might be nice to add a plot of the winds as supplemental material (even if it is just one sonde above the source) and discuss if the back azimuth deviations are consistent with the cross winds of the specified propagation duct.

RESPONSE: A multiple plot of the crosswind situation is added to the supplement for each of the four stations and a profile along the source to the station. It should allow insights on the crosswind situation and possible resulting backazimuth deviations along the complete signal propagation path. Nevertheless the situation is not that clear and simple (so that a single crosswind plot above the source could not provide a realistic representation), since atmospheric profiles, also those of the crosswind, change with range and altitude. It is not clear and easy to say which altitude parts influence the signals most and lead to back-azimuth variations. Some considerations are added to the caption, but they cannot cover the whole picture of a changing atmosphere along thousands of kilometers. It can be reconsidered if this additional picture is worth to be shown. The complementary figure is referenced in the given paragraph (from line 202) where back-azimuth deviations are mentioned.

Line 241:
This sentence is unnecessary and could be removed.

RESPONSE: Agreed, the sentence was removed from the manuscript.

Line 243:
Please clarify the phrase "acoustic velocities". Do you mean apparent velocity?

RESPONSE: Yes, "apparent" was added to the text.

Line 244:

It should be qualified that celerities outside these bounds exclude purely acoustic waves from the origin at the time of rupture.

RESPONSE: We modified the phrase highlighting origin and rupture time, according to the reviewer's suggestions.

Lines 260-265:
Please be more clear on your definitions of both attenuation and propagation modeling. You calculate them using different methods, but they both show results of transmission loss with your attenuation shown in map view and the propagation modeling shown as a cross section.

RESPONSE: According to the comments of both reviewer's we streamlined the language around the concept of transmission loss and skipped the misleading use of "attenuation modeling". We thus define the semi-empirical technique for transmission loss calculations (fig3) as the first technique used, its description modified in various text passages. We keep the concept of propagation modeling (fig4) as a second technique and clarified the differences and benefits of those two concepts both applied in the study. Lines 260-265 are modified to: "Transmission loss calculations using firstly a semi-empirical method for a horizontal representation (map view, figure 3) and secondly a parabolic-equation-based propagation model for a vertical representation (cross section, figure 4) are performed...".

Line 268:
What is the spatial resolution of the range dependent atmospheric profiles? 0.5 x 0.5 degree as stated in Figure 3 caption? Please state this in the manuscript text.

RESPONSE: the 0.5° x 0.5° resolution from the figure caption is also added to the manuscript text here.

Line 270:
Please elaborate on what INFERNO is and how you use it.

RESPONSE: The INFERNO software, its key features and its application within this study is elaborated in more detail in the given passage, following the recommendation of both reviewers.

Lines 266-295:
The transmission losses, associated uncertainties for each array, and source frequency in this paragraph could all be put into a table to more concise and easily get your point across to the reader.

RESPONSE: A table was added according to the reviewer's suggestion and the according manuscript text was modified; the numbers were shifted to the table and the text was shortened to just give ranges of transmission loss values and uncertainties.

Figure 4:
Why did you choose 1Hz modeling? That seems a bit high for propagation of such large distances.

RESPONSE: This is true, but 1 Hz is a compromise between the 0.2 Hz (figure 3a) of the low-frequency acoustic energy maximum and the 3 Hz (figure 3b) of the high-frequency signal content which is unique for the stratospheric ducted signal parts. It is also a matter of necessity to be able to present the stratospheric ducts in a clear and concise figure. When going to lower frequencies, there is less attenuation and higher wavelengths and the modeling as well as picture becomes less detailed and more blurred, losing the clear optical representation of the stratospheric waveguides which are the main message of the figure.

Figure 5:
Lines 384-395 refer to Sulawesi Island, North Maluku, etc. It might be helpful to label these areas in the figure so the reader can more easily follow along. Or, if you chose to add an inset to Figure 1 you can refer to the locations there.

RESPONSE: References to island names were added as labels to figure 1 and 5, instead on an inset, a second subfigure is added to figure 1 (see separate comment).

Line 425:
Tiny semantic issue, but I don't believe the earthquake is named after the mountain. The earthquake ruptured the Denali fault as well as the Toschunda fault and is located in/near Denali National Park.

RESPONSE: Good to know, we changed the mountain association to the fault one.

Line 449:
Do you have a reference for this?

RESPONSE: Unfortunately not. To my knowledge there was no infrasound study about these events yet, so the statements "are estimations following data analyses performed by authors of this study" (this is already mentioned in the manuscript).

Line 463:
I would argue that both topography and conducive propagation conditions are necessary for detection.

RESPONSE: Yes, the "conducive propagation conditions" are added to the end of the paragraph, where station detection and not source characteristics are described.

**Technical Corrections:**

Line 25
"is supposed to" sounds a bit awkward
Perhaps "The seismic-to-acoustic coupling at nearby terrain features *is shown to* generate distinct infrasonic signatures clearly recordable at remote infrasound arrays.

RESPONSE: instead of a correction of the wording the complete sentence was omitted since it is more or less redundant to the rest of the abstract and critical in the context of the reviewer1 nomenclature harmonization comment (also reviewer2 comment to line 79 and in general).

Line 27:
Suggest adding the word "infrasound" for clarity.
Event-related *infrasound* observations

RESPONSE: the phrase was modified according to the reviewer's suggestion.

Lines 30, 95, 148, 358:
In these lines, "back tracking" is used. Please choose one term (I think "back projection" is the most widely used) and use throughout the manuscript.

RESPONSE: the nomenclature was modified to a homogenized use of "back projection" throughout the manuscript according to both reviewers' suggestions.

Line 41:
Do you mean "a very high *rate of* natural seismicity"?

RESPONSE: the phrase was modified according to the reviewer's suggestion.

Line 63:
"in the course of" may better be stated as "surrounding this event"

RESPONSE: the phrase was modified according to the reviewer's suggestion.

Line 65:
Would it be more appropriate to say "both" instead of "either"?

RESPONSE: the phrase was modified according to the reviewer's suggestion.

Line 179:
I don't think "from above" is necessary.

RESPONSE: the phrase was skipped according to the reviewer's suggestion.

Line 196:
There should be a better way to cite this. Please check the NHESS citation guidelines.

RESPONSE: there is a USGS event page to the Sulawesi 2018 earthquake, which we referenced according to the reviewer's suggestion.

Figure 3:
The stratospheric wind vectors pretty hard to see. Can you make them slightly larger

RESPONSE: Arrow length and arrow head size of stratospheric wind vectors in figure 3 were enlarged according to the reviewer's suggestion.

[revised manuscript text omitted]